# TRAINED MODELS TELL US HOW TO MAKE THEM ROBUST TO SPURIOUS CORRELATION WITHOUT GROUP ANNOTATION

## ABSTRACT

Classifiers trained with Empirical Risk Minimization (ERM) tend to rely on attributes that have high spurious correlation with the target. This can degrade the performance on underrepresented (or *minority*) groups that lack these attributes, posing significant challenges for both out-of-distribution generalization and fairness objectives. Many studies aim to enhance robustness to spurious correlation, but they sometimes depend on group annotations for training. Additionally, a common limitation in previous research is the reliance on group-annotated validation datasets for model selection. This constrains their applicability in situations where the nature of the spurious correlation is not known, or when group labels for certain spurious attributes are not available. To enhance model robustness with minimal group annotation assumptions, we propose Environment-based Validation and Loss-based Sampling (EVaLS). It uses the losses from an ERM-trained model to construct a balanced dataset of high-loss and low-loss samples, mitigating group imbalance in data. This significantly enhances robustness to group shifts when equipped with a simple post-training last layer retraining. By using environment inference methods to create diverse environments with correlation shifts, EVaLS can potentially eliminate the need for group annotation in validation data. In this context, the worst environment accuracy acts as a reliable surrogate throughout the retraining process for tuning hyperparameters and finding a model that performs well across diverse group shifts. EVaLS effectively achieves group robustness, showing that group annotation is not necessary even for validation. It is a fast, straightforward, and effective approach that reaches near-optimal worst group accuracy without needing group annotations, marking a new chapter in the robustness of trained models against spurious correlation.

## 1 INTRODUCTION

Training deep learning models using Empirical Risk Minimization (ERM) on a dataset, poses the risk of relying on *spurious correlation*. These are correlations between certain patterns in the training dataset and the target (e.g., the class label in a classification task) despite lacking any causal relationship. Learning such correlations as shortcuts can negatively impact the models' accuracy on *minority groups* that do not contain the spurious patterns associated with the target (Kirichenko et al., 2023; LaBonte et al., 2023). This problem leads to concerns regarding fairness (Hashimoto et al., 2018), and can also cause a marked reduction in the performance. This occurs particularly when minority groups, which are underrepresented during training, become overrepresented at the inference time, as a result of shifts within the subpopulations (Yang et al., 2023b). Hence, ensuring robustness to group shifts and developing methods that improve *worst group accuracy* (WGA) is crucial for achieving both fairness and robustness in the realm of deep learning.

Many studies have proposed solutions to address this challenge. A promising line of research focuses on increasing the contribution of minority groups in the model's training (Liu et al., 2021a; Yang et al., 2023a; Sagawa et al., 2019). A strong assumption that is considered by some previous works is having access to group annotations for training or fully/partially fine-tuning a pretrained model (Nam et al., 2021; Sagawa et al., 2019; Kirichenko et al., 2023). The study by Kirichenko et al. (2023) proposes that retraining the last layer of a model on a dataset that is balanced in terms

of group annotation can effectively enhance the model's robustness against shifts in spurious correlation. While these works have shown tremendous robustness performance, their assumption for the availability of the group annotation restricts their usage.

In many real-world applications, the process of labeling samples according to their respective groups can be prohibitively expensive, and sometimes impractical, especially when all minority groups may not be identifiable beforehand. A widely adopted strategy in these situations involves the indirect inference of various groups, followed by the training of models using a loss function that is balanced across groups (Liu et al., 2021a; Qiu et al., 2023; Nam et al., 2020; Yang et al., 2023b). The loss value of the model, or its alternatives, are popular signals for recognizing minority groups (Liu et al., 2021a; Qiu et al., 2023; Nam et al., 2020; Noohdani et al., 2024). While most of these techniques necessitate full training of a model, Qiu et al. (2023) attempt to adapt the DFR method (Kirichenko et al., 2023) with the aim of preserving computational efficiency while simultaneously improving robustness to the group shift. However, this method still requires group annotations of the validation set for the model selection and hyperparameter tuning. Consequently, this constitutes a restrictive assumption when adequate annotations for certain groups are not supplied. It also applies to situations where some shortcut attributes are completely unknown.

In this study, we present a novel strategy that effectively mitigates reliance on spurious correlation, completely eliminating the need for group annotations during both training and retraining. More interestingly, we provide empirical evidence indicating that group annotations are not necessary, even for model selection. We show that assembling a diverse collection of environments for model selection, which reflects group shifts can serve as an effective alternative approach. Our proposed scheme, Environment-based Validation and Loss-based Sampling (EVaLS), strengthens the robustness of trained models against spurious correlation, all without relying on group annotations. EVaLS is pioneering in its ability to eliminate the need for group annotations at *every phase*, including the model selection step. EVaLS posits that in the absence of group annotations, a set of *environments* showcasing group shifts is sufficient. Worst Environment Accuracy (WEA) could then be utilized for model selection. We observe that spurious correlations, as a form of subpopulation shifts, cause significant group shifts when using environment inference methods (Creager et al., 2021). Consequently, the inferred environments—which could be obtained even by simply dividing validation data based on predictions from a random linear layer atop a trained model's feature space—can effectively compare different sets of hyperparameters for tuning. Figure 1 demonstrates the overall procedure of the main parts of EVaLS.

Aligned with AFR (Qiu et al., 2023) and DFR (Kirichenko et al., 2023), EVaLS offers a significant advantage by not requiring any modifications to the standard ERM training procedure or the original training data. Moreover, it does not require information from the initial phases of ERM training, such as an early-stopped model. This characteristic is particularly beneficial in enhancing the robustness of ERM-pretrained networks against their potential inherent biases. Specifically, it eliminates the need to retrain the entire model, which may be impractical or infeasible when the original training data is unavailable.

Our empirical observations support prior research which suggests that high-loss data points in a trained model may signal the presence of minority groups (Liu et al., 2021a; Qiu et al., 2023; Nam et al., 2020). EVaLS evenly selects from both high-loss and low-loss data to form a balanced dataset that is used for last-layer retraining. We offer theoretical explanations for the effectiveness of this approach in addressing group imbalances, and experimentally show the superiority of our efficient solution to the previous strategies. Comprehensive experiments conducted on spurious correlation benchmarks such as CelebA (Liu et al., 2014), Waterbirds (Sagawa et al., 2019), and UrbanCars (Li et al., 2023), demonstrate that EVaLS achieves optimal accuracy. Moreover, when group annotations are accessible solely for model selection, our approach, EVaLS-GL, exhibits enhanced performance against various distribution shifts, including attribute imbalance, as seen in MultiNLI (Williams et al., 2017), and class imbalance, exemplified by CivilComments (Borkan et al., 2019). We further present a new dataset, *Dominoes Colored-MNIST-FashionMNIST*, which depicts a situation featuring multiple independent shortcuts, that group annotations are only available for part of them (see Section 2.2). In this setting, we show that strategies with lower levels of group supervision are paradoxically more effective in mitigating the reliance on both known and unknown shortcuts.

The main contributions of this paper are summarized as follows:

- We present EVaLS, a simple yet effective post-hoc approach that enhances the robustness of ERM-pretrained models against both known and unknown spurious correlations, without relying on ground-truth group annotations.
- We offer both theoretical and empirical insights on how balanced sampling from high-loss and low-loss samples offers a dataset in which the group imbalance is notably mitigated.
- Using simple environment inference techniques, EVaLS introduces worst environment accuracy as a reliable indicator for model selection.
- EVaLS achieves near-optimal performance in spurious correlation benchmarks with zero group annotations, and delivers state-of-the-art performance when group annotations are available for model selection.
- By utilizing a newly introduced dataset with two spurious attributes, we demonstrate that EVaLS improves robustness to both known and unknown spurious attributes learned by an ERM-trained model better than methods relying on group information.

## 2 PRELIMINARIES

### 2.1 PROBLEM SETTING

We assume a general setting of a supervised learning problem with distinct data partitions $\mathcal{D}^{\mathrm{Tr}}$ for training, $\mathcal{D}^{\mathrm{Val}}$ for validation, and $\mathcal{D}^{\mathrm{Te}}$ for final evaluation. Each dataset comprises a set of paired samples $(x, y)$, where $x \in \mathcal{X}$ represents the data and $y \in \mathcal{Y}$ denotes the corresponding labels. Conventionally, $\mathcal{D}^{\mathrm{Tr}}$, $\mathcal{D}^{\mathrm{Val}}$, and $\mathcal{D}^{\mathrm{Te}}$ are assumed to be uniformly sampled from the same distribution. However, this idealized assumption does not hold in many real-world problems where distribution shift is inevitable. In this context, we consider the subpopulation shift problem (Yang et al., 2023b). In a general form of this setting, it is assumed that data samples consist of different groups $\mathcal{G}_i$, where each group comprises samples that share a property. More specifically, the overall data distribution $p(x, y) = \sum_i \alpha_i p_i(x, y)$ is a composition of individual group distributions $p_i(x, y)$ weighted by their respective proportions $\alpha_i$, where $\sum_i \alpha_i = 1$. In this work, we assume that $\mathcal{D}^{\mathrm{Tr}}$, $\mathcal{D}^{\mathrm{Val}}$, and $\mathcal{D}^{\mathrm{Te}}$ are composed of identical groups but with a different set of mixing coefficients $\{\alpha_i\}$. It is noteworthy that the validation set may have approximately identical coefficients to those of the training or testing sets, or it may have entirely different coefficients.

Several kinds of subpopulation shifts are defined in the literature, including class imbalance, attribute imbalance, and spurious correlation (Yang et al., 2023b). Class imbalance refers to the cases where there is a difference between the proportion of samples from each class, while attribute imbalance occurs when instances with a certain attribute are underrepresented in the training data, even though this attribute may not necessarily be a reliable predictor of the label. On the other hand, spurious correlation occurs when various groups are differentiated by spurious attributes that are partially predictive and correlated with class labels but are causally irrelevant. More precisely, we can consider a set of spurious attributes $\mathcal{S}$ that partition the data into $|\mathcal{S}| \times |\mathcal{Y}|$ groups. When the concurrence of a spurious attribute with a label is significantly higher than its correlation with other labels, that spurious attribute could become predictive of the label, resulting in deep models relying on the spurious attributes as shortcuts instead of the core ones. This is followed by a decrease in the model's performance on groups that do not have this attribute.

Given a class, the group containing samples with correlated spurious attributes is referred to as *majority* group of that class, while the other groups are called the *minority* groups. As an example, in the Waterbirds dataset (Sagawa et al., 2019), for which the task is to classify images of birds into landbird and waterbird, there are spurious attributes {*water background*, *land background*}. Each background is spuriously correlated with its associated label, decompose the data into two majority groups *waterbird on water background*, and *landbird on land background*, and two minority groups *waterbird on land background* and *landbird on water background*. Our goal is to make the classifier robust to spurious attributes by increasing performance for all groups.

### 2.2 ROBUSTNESS OF A TRAINED MODEL TO UNKNOWN SHORTCUTS

In scenarios where group annotations are absent, traditional methods that depend on these annotations for training or model selection become infeasible. Moreover, as previously discussed by Li

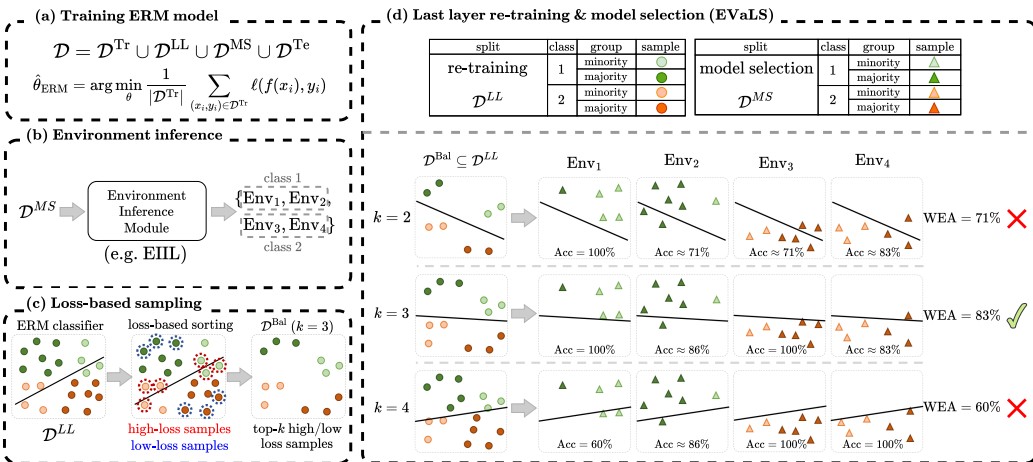

Figure 1: Overview of the proposed approach. (a) We randomly split the dataset $\mathcal{D}$ into $\mathcal{D}^{\text{Tr}}$, $\mathcal{D}^{\text{MS}}$, $\mathcal{D}^{\text{LL}}$ and $\mathcal{D}^{\text{Te}}$. We train the initial classifier on $\mathcal{D}^{\text{Tr}}$ with empirical risk minimization (ERM). Alternatively, we can assume that an ERM-trained model is given. (b) An environment inference method is utilized to infer diverse environments for each class of $\mathcal{D}^{\text{MS}}$. (c) We evaluate $\mathcal{D}^{\text{LL}}$ samples on the initial ERM classifier and sort high-loss and low-loss samples of each class for loss-based sampling. (d) Finally, we perform last-layer retraining on the loss-based selected samples $\mathcal{D}^{\text{Bal}}$. Each retraining setting (e.g. different $k$ for loss-based sampling) is validated based on the worst accuracy of the inferred environments. Note that majority and minority groups are shown with dark and light colors for better visualization, but are not known in our setting.

et al. (2023), when data contains multiple spurious attributes and annotations are only available for some of them, such methods would make the model robust only to the known spurious attributes. To further explore such complex scenarios, we introduce the *Dominoes Colored-MNIST-FashionMNIST (Dominoes CMF)* dataset (Figure 4(a)). Drawing inspiration from Pagliardini et al. (2022a) and Arjovsky et al. (2020), Dominoes CMF merges an image from CIFAR10 (Krizhevsky & Hinton, 2009) at the top with a colored (red or green) MNIST (Deng, 2012) or FashionMNIST (Xiao et al., 2017) image at the bottom. The primary label is derived from the CIFAR10 image, while the bottom part introduces two independent spurious attributes: color (red or green) and style (MNIST or FashionMNIST). Although annotations for shape are provided for training and model selection, color remains an unknown variable until testing. For more details on the dataset refer to the Appendix.

The illustrations in Figure 2(a-c) depict the outlined scenario. A classifier trained using ERM is dependent on both spurious features (Figure 2(b)). Yet, achieving robustness against one spurious correlation (Figure 2(c)), does not ensure robustness against both (Figure 2(a)). In Section 4 we show that our approach, which does not rely on the group annotations of the identified group, achieves enhanced robustness to both spurious correlations, outperforming strategies that depend on the known group's information.

## 3 ENVIRONMENT-BASED VALIDATION AND LOSS-BASED SAMPLING

EVaLS is designed to improve the robustness of ERM-trained deep learning models to group shifts without the need for group annotation. In line with the DFR (Kirichenko et al., 2023) approach, we utilize a classifier defined as $f = h_\phi \circ g_\theta$, where $g_\theta$ represents a deep neural network serving as a feature extractor, and $h_\phi$ denotes a linear classifier. The classifier is initially trained with the ERM objective on the training dataset $\mathcal{D}^{\text{Tr}}$. Subsequently, we freeze the feature extractor $g_\theta$ and focus solely on retraining the last linear layer $h_\phi$ using the validation dataset $\mathcal{D}^{\text{Val}}$ as a held-out dataset. This scheme helps us make our method available in settings where $\mathcal{D}^{\text{Tr}}$ is not available, or where repeating the training process is infeasible.

We randomly divide the validation set $\mathcal{D}^{\text{Val}}$ into two subsets, $\mathcal{D}^{\text{LL}}$ and $\mathcal{D}^{\text{MS}}$ which are used for last layer training and model selection, respectively. In Section 3.1 we explain how to sample a

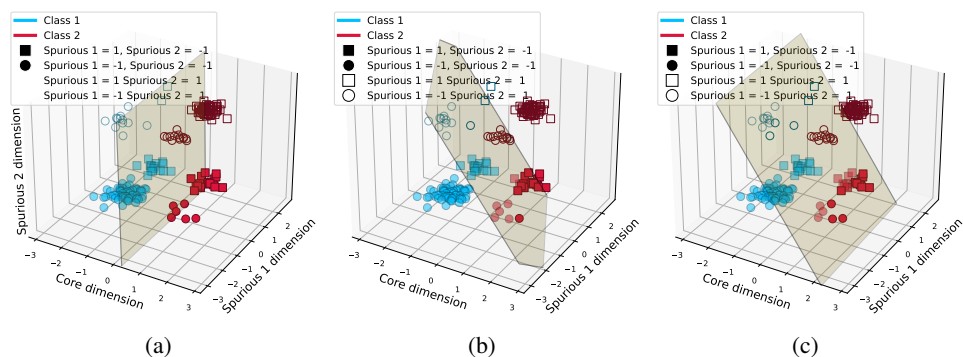

(a)          (b)          (c)

Figure 2: (a) If all spurious attributes in a dataset are known, they can be utilized to fit a classifier that captures the essential attributes. (b) In the absence of knowledge about all spurious attributes, the model would depend on them for classification, leading to incorrect classification of minority samples. (c) If some spurious attribute is unknown (Spurious 2), the model becomes robust only to the known spurious correlations (Spurious 1), but it still underperforms on minority samples.

subset of $\mathcal{D}^{LL}$ that statistically handles the group shifts inherent in the dataset. In Section 3.2 we describe how $\mathcal{D}^{MS}$ is divided into different environments that are later used for model selection. The optimal number of selected samples from $\mathcal{D}^{LL}$ and other hyperparameters is determined based on the worst environment accuracies among environments that are obtained from $\mathcal{D}^{MS}$. By combining our sampling and validation strategy, we aim to provide a robust linear classifier $h_{\phi^*}$ that significantly improves the accuracy of underrepresented groups without requiring group annotations of training or validation sets. Finally in Section 3.3, we provide theoretical support for the loss-based sampling procedure and its effectiveness. Figure 1 illustrates the comprehensive workflow of the EVaLS.

### 3.1 LOSS-BASED INSTANCE SAMPLING

Following previous works (Liu et al., 2021a; Nam et al., 2020; Qiu et al., 2023), we use the loss value as an indicator for identifying minority groups. We first evaluate classifier $f$ on samples within $\mathcal{D}^{LL}$ and choose $k$ samples with the highest and lowest loss values in each class for a given $k$. By combining these $2k$ samples from each class, we construct a balanced set $\mathcal{D}^{Bal}$, consisting of high-loss and low-loss samples (see Figure 1(c)). $\mathcal{D}^{Bal}$ is then used for the training of the last layer of the model. As depicted in Figure 3, the proportion of minority samples among various percentiles of samples with the highest loss values increases as we select a smaller subset of samples with the highest loss. This suggests that high and low-loss samples could serve as effective representatives of minority and majority groups, respectively. In Section 3.3, we offer theoretical insights explaining why this approach could lead to the creation of group-balanced data.

### 3.2 PARTITIONING VALIDATION SET INTO ENVIRONMENTS

Contrary to common assumptions and practices in the field, precise group labels for the validation set are not essential for training models robust to spurious correlations. Our empirical findings, detailed in Section 4, reveal that partitioning the validation set into environments that exhibit significant subpopulation shifts can be used for model selection. Under these conditions, the worst environment accuracy (WEA) emerges as a viable metric for selecting the most effective model and hyperparameters.

The concept of an *environment*, as frequently discussed in the invariant learning literature, denotes partitions of data that exhibit different distributions. A model that consistently excels across these varied environments, achieving impressive worst environment accuracy (WEA), is likely to perform equally well across different groups in the test set. Several methods for inferring environments with notable distribution shifts have been introduced (Creager et al., 2021; Liu et al., 2021b). Environment Inference for Invariant Learning (EIIL) (Creager et al., 2021), leverages the predictions from an earlier trained ERM model to divide the data into two distinct environments that significantly deviate from the invariant learning principle proposed by Arjovsky et al. (2020), thus creating en-

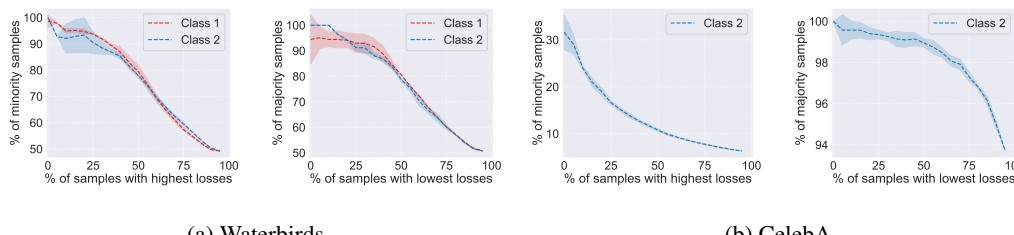

(a) Waterbirds           (b) CelebA

Figure 3: The percentage of samples with the highest (lowest) losses across various thresholds that belong to the minority (majority) group within different classes in $\mathcal{D}^{\text{LL}}$ for (a) the Waterbirds and (b) CelebA datasets. Minority group samples are more prevalent among high-loss samples, while majority group samples dominate the low-loss areas. The error bars are calculated across three ERM models. [1]

vironments with distribution shifts. Initially, EIIL is employed to split $\mathcal{D}^{\text{MS}}$ into two environments. Subsequently, each environment is further divided based on sample labels, resulting in $2 \times |\mathcal{Y}|$ environments. To measure the difference between the distribution of environments, we define *group shift* of a class as the absolute difference in the proportion of a minority group between two environments of that class. A higher group shift suggests a more distinct separation between environments. As detailed in the Appendix, environments inferred by EIIL demonstrate an average group shift of 28.7% over datasets with spurious correlation. Further information about EIIL and the group shift quantities for each dataset can be found in the Appendix.

We demonstrate that even more straightforward techniques, such as applying a random linear layer over the feature embedding space and distinguishing environments based on correctly and incorrectly classified samples of each class, can be effective to an extent in several cases (See Appendix F.3). It underscores that the feature space of a trained model is a valuable resource of information for identifying groups affected by spurious correlations. This supports the logic of previous research that employs clustering (Sohoni et al., 2020) or contrastive methods (Zhang et al., 2021) in this space to differentiate between groups.

### 3.3 THEORETICAL ANALYSIS

The environments obtained as described in Section 3.2 are utilized for hyperparameter tuning, specifically for tuning $k$, which is the number of selected samples from loss tails. It is known that minority samples are more prevalent among high-loss samples, while majority samples dominate the low-loss category. However, the question remains whether loss-based sampling can construct a balanced dataset without introducing spurious correlations. In this section, aligned with our practical approach, we provide theoretical insights into how loss-based sampling within a class can be used to create a group-balanced dataset.

Consider a binary classification problem with a cross-entropy loss function. Let logits be denoted as $L$. We assume a general assumption that in feature space (output of $g_\theta$) samples from the minority and majority of a class are derived from Gaussian distributions. As a result, we can consider $\mathcal{N}(\mu_{\min}, \sigma_{\min}^2)$ and $\mathcal{N}(\mu_{\text{maj}}, \sigma_{\text{maj}}^2)$ as the distribution of minority and majority samples in logits space (See Lemma D.1 in Appendix D for details). Because the loss function is a monotonic function of logits, the tails of the distribution of loss across samples are equivalent to that of the logits in each class.

**Proposition 3.1.** *[Feasiblity Of Loss-based Group Balancing] Suppose that $L$ is derived from the mixture of two distributions $\mathcal{N}(\mu_{min}, \sigma_{min}^2)$ and $\mathcal{N}(\mu_{maj}, \sigma_{maj}^2)$ with proportion of $\varepsilon$ and $1 - \varepsilon$, respectively, where $\varepsilon \leq \frac{1}{2}$. If (i) $\sigma_{min} > \sigma_{maj}$, or (ii) under sufficient and necessary conditions on $\mu_{min}, \mu_{maj}, \sigma_{min}$ and $\sigma_{maj}$ including inequality 1 (see App.D), there exists $\alpha$ and $\beta$ such that restricting $L$ to the $\alpha$-left and $\beta$-right tails of its distribution results in a group-balanced distribution; in which both components are equally represented.*

---

[1] Note that in the CelebA dataset, only the "blond hair" class includes a minority group.

$$\epsilon \geq \text{sigmoid}\left( -\frac{(\mu_{\text{maj}} - \mu_{\text{min}})^2}{2(\sigma_{\text{maj}}^2 - \sigma_{\text{min}}^2)} - \log\left(\frac{\sigma_{\text{maj}}}{\sigma_{\text{min}}}\right) \right) \qquad (1)$$

We provide an outline for proof of Proposition 3.1 here and leave the complete and formal proof and also exact bounds to Appendix D. We also analyze the conditions and effects of spurious correlation in satisfying these conditions. Practical justifications for Proposition 3.1 can be found in Appendix D.2. To proceed with the outline, we first define a key concept.

**Definition 3.1** (Proportional Density Difference). *For any interval $I = (a, b]$ and a mixture distribution $\varepsilon P_1(x) + (1 - \varepsilon)P_2(x)$, the proportional density difference is defined as the difference of accumulation of two component distributions in the interval $I$ and is denoted by $\Delta_\varepsilon P_{mixture}(I)$.*

$$\Delta_\varepsilon P_{mixture}(I) \triangleq \varepsilon P_1\big(x \in I\big) - (1 - \varepsilon)P_2\big(x \in I\big) \qquad (2)$$

**Proof outline**    Our proof proceeds with three steps. First, we reformulate the theorem as an equality of left- and right-tail proportional distribution differences. In other words, we show that the more mass the minority distribution has on one tail, the more mass the majority distribution must have on the other tail. Afterward, supposing $\mu_{\text{min}} < \mu_{\text{maj}}$ WOLG, we propose a proper range for $\beta$ values on the right tail. We show that when $\sigma_{\text{maj}} \leq \sigma_{\text{min}}$, values for $\alpha$ trivially exist that can overcome the imbalance between the two distributions. In the last step, for the case in which the variance of the majority is higher than the minority, we discuss a necessary and sufficient condition for the existence of $\alpha$ and $\beta$ based on the left-tail proportional density difference using the properties of its derivative with respect to $\alpha$.

Condition 1 suggests that for a given degree of spurious correlation $\epsilon$ and variations $\sigma_{\text{maj}}, \sigma_{\text{min}}$, an essential prerequisite for the efficacy of loss-based sampling is a sufficiently large disparity between the mean distributions of minority and majority samples, denoted by $\|\mu_{\text{maj}} - \mu_{\text{min}}\|^2$. This indicates that the groups should be distinctly separable in the logits space.

Although the parameters $\alpha$ and $\beta$ are theoretically established under certain conditions, their actual values remain undetermined. Therefore, validation data is essential to identify the appropriate tails. For practicality and simplicity, we assume an equal number $k$ of samples for both tails and explore this count (high- and low-loss samples) from a predefined set of values. By leveraging the worst environment accuracy on validation data after last-layer retraining, as detailed in Section 3.2, we identify the optimal candidate that ensures uniform accuracy across all environments.

## 4 EXPERIMENTS

In this section, we evaluate the effectiveness of the proposed scheme through comprehensive experiments on multiple datasets and compare it with various methods and baselines. We begin by briefly describing evaluation datasets and then introduce baselines and comparative methods. Finally, we report and fully explain the results.

**Datasets**    Our approach, along with other baselines, is evaluated on Waterbirds (Sagawa et al., 2019), CelebA (Liu et al., 2014), UrbanCars (Li et al., 2023), CivilComments (Borkan et al., 2019), and MultiNLI (Williams et al., 2017). As per the study by Yang et al. (2023b), Waterbirds, CelebA, and UrbanCars among these datasets exhibit spurious correlation. Among the rest, CivilComments has class and attribute imbalance, whereas MultiNLI exhibits attribute imbalance. For additional details on the datasets, please refer to the Appendix E.3.

**Baselines**    We compare EVaLS with six baselines in addition to standard ERM. **Group-DRO** (Sagawa et al., 2019) trains a model on the data with the objective of minimizing its average loss on the minority samples. This method requires group labels of both the training and validation sets. **DFR** (Kirichenko et al., 2023) argues that models trained with ERM are capable of extracting the core features of images. Thus, it first trains a model with ERM, and retrains only the last linear classifier layer on a group-balanced subset of the validation or the held-out training data. While DFR reduces the number of group-annotated samples, it still requires group labels in the training phase. **GroupDRO + EIIL** (Creager et al., 2021) infers environments of the training set and trains a model

with GroupDRO on the inferred environments. **JTT** (Liu et al., 2021a) first trains a model with ERM on the dataset, and then retrains it on the dataset by upweighting the samples that were misclassified by the initial ERM model. **ES Disagreement SELF** (LaBonte et al., 2023) selects samples with the highest difference in output when comparing an ERM-trained model to its early-stopped version. Then, they fine-tune the last layer of the ERM-trained model on the selected samples. **AFR** (Qiu et al., 2023) trains a model with standard ERM, and retrains the classifier on a weighted held-out data. The weights assigned to retraining samples are determined by the probability that the ERM-pretrained model assigns to the ground-truth label, leading to an increased weighting of samples from minority groups.

GroupDRO + EIIL, JTT, ES Disagreement SELF, and AFR eliminate the reliance on group annotations for their (re)training. However, unlike EVaLS, they all require group labels for model selection. JTT, GroupDRO, and GroupDRO + EIIL necessitate training the entire model to apply their methods. Additionally, ES Disagreement and SELF require early-stopped versions during training with ERM. In contrast, DFR, AFR, and EVaLS operate in a completely post-training manner without relying on any information from ERM training. This property makes these methods applicable in real-world scenarios when training checkpoints or training data are unavailable, or when it is infeasible to repeat the training due to reasons such as a large training set.

**Setup** Similar to all the works mentioned in Section 4, we use ResNet-50 (He et al., 2016) pretrained on ImageNet (Russakovsky et al., 2015) for image classification tasks. We used random crop and random horizontal flip as data augmentation, similar to Kirichenko et al. (2023). For a fair comparison with the baselines, we did not employ any data augmentation techniques in the process of retraining the last layer of the model. For the CivilComments and MultiNLI, we use pretrained BERT (Devlin et al., 2019) and crop sentences to 220 tokens length. In EvaLS, we use the implementation of EIIL by `spuco` package (Joshi et al., 2023) for environments inference on the model selection set with 20000 steps, SGD optimizer, and learning rate $10^{-2}$ for all datasets.

Model selection and hyper-parameter fine-tuning are done according to the worst environment (or group if annotations are assumed to be available) accuracy on the validation set. For each dataset, we assess the performance of our model in two cases: fine-tuning the ERM classifier or retraining it. For all datasets except MultiNLI and Urbancars, retraining yielded better validation results. We report the results of our experiments in two settings: (i) EVaLS, which incorporates loss-based instance sampling for training the last layer, and environment inference for model selection. (ii) EVaLS-GL, similar to EVaLS except in using ground-truth group labels for model selection. For more details on the ERM training and last layer re-training hyperparameters refer to the Appendix.

## 4.1 RESULTS

The results of our experiments along with the reported results on GroupDRO (Sagawa et al., 2019), DFR (Kirichenko et al., 2023), JTT (Liu et al., 2021a), ES Disagreement SELF (LaBonte et al., 2023), and AFR (Qiu et al., 2023) on five datasets are shown in Table 1. The reported results for GroupDRO, DFR, JTT, and AFR except those for the UrbanCars are taken from Qiu et al. (2023). For EIIL+Group DRO, the results for Waterbirds, CelebA, and CivilComments are reported from Zhang et al. (2021). The results of SELF on CelebA and MultiNLI are reported from the original paper (LaBonte et al., 2023). We report only the worst group accuracy of methods in Table 1. The average group accuracies are documented in the Appendix. The Group Info column shows whether group annotation is required for training or model selection entry for each method. Methods that do not require information regarding ERM training (such as training data or checkpoints) are identified with a $star$ in the table.

Overall, our approaches outperform methods that do not require group annotations for (re)training in 2 out of 3 datasets with spurious correlations. Moreover, EVaLS-GL surpasses other methods with a similar level of group supervision on MultiNLI (Williams et al., 2017) and achieves state-of-the-art performance among all methods on UrbanCars (Li et al., 2023). Furthermore, EVaLS and EVaLS-GL, similar to DFR (Kirichenko et al., 2023) and AFR (Qiu et al., 2023), can be applied to ERM-trained models without needing further information about their training.

Table 1: Comparison of worst group accuracy across various methods, including ours, on five datasets. The Group Info column indicates if each method utilizes group labels of the training/validation data, with ✓✓ denoting that group information is employed during both stages. Bold numbers are the highest results overall, while underlined ones are the best among methods that may require group annotation only for model selection. CivilComments is class imbalanced, MultiNLI has imbalanced attributes, and the other three datasets have spurious correlations. The × sign indicates that the dataset is out of the scope of the method. Methods that do not rely on ERM training information are identified with ⋆. Mean and standard deviation are calculated over three runs.

| Method | Group Info | Datasets | | | | |
|---|---|---|---|---|---|---|
| | Train/Val | Waterbirds | CelebA | UrbanCars | CivilComments | MultiNLI |
| GDRO (Sagawa et al., 2019) | ✓/✓ | 91.4 | **88.9** | 73.1 | 69.9 | **77.7** |
| DFR⋆ (Kirichenko et al., 2023) | ✗/✓✓ | **92.9**$_{\pm 0.2}$ | 88.3$_{\pm 1.1}$ | 79.6$_{\pm 2.2}$ | **70.1**$_{\pm 0.8}$ | 74.7$_{\pm 0.7}$ |
| GDRO + EIIL (Creager et al., 2021) | ✗/✓ | 77.2$_{\pm 1}$ | 81.7$_{\pm 0.8}$ | 76.5$_{\pm 2.6}$ | 67.0$_{\pm 2.4}$ | 61.2$_{\pm 0.5}$ |
| JTT (Liu et al., 2021a) | ✗/✓ | 86.7 | 81.1 | 79.5 | 69.3 | 72.6 |
| SELF (LaBonte et al., 2023) | ✗/✓ | 91.6$_{\pm 1.4}$ | 83.9$_{\pm 0.9}$ | 83.2$_{\pm 0.8}$ | 66.0$_{\pm 1.7}$ | 70.7$_{\pm 2.5}$ |
| AFR⋆ (Qiu et al., 2023) | ✗/✓ | 90.4$_{\pm 1.1}$ | 82.0$_{\pm 0.5}$ | 80.2$_{\pm 2.0}$ | 68.7$_{\pm 0.6}$ | 73.4$_{\pm 0.6}$ |
| EVaLS-GL⋆ (Ours) | ✗/✓ | 89.4$_{\pm 0.3}$ | 84.6$_{\pm 1.6}$ | **83.5**$_{\pm 1.7}$ | 68.0$_{\pm 0.5}$ | 75.1$_{\pm 1.2}$ |
| EVaLS⋆ (Ours) | ✗/✗ | 88.4$_{\pm 3.1}$ | 85.3$_{\pm 0.4}$ | 82.1$_{\pm 0.9}$ | × | × |
| ERM | ✗/✗ | 66.4$_{\pm 2.3}$ | 47.4$_{\pm 2.3}$ | 18.67$_{\pm 2.0}$ | 61.2$_{\pm 3.6}$ | 64.8$_{\pm 1.9}$ |

The comparison between EVaLS and GroupDRO + EIIL indicates that when environments are available instead of groups, our method, which uses environments solely for model selection and utilizes loss-based sampling, is more effective than GroupDRO, a potent invariant learning method.

Regarding the UrbanCars, which contains an un-annotated spurious attribute, Li et al. (2023) has shown that shortcut mitigation methods often struggle to address multiple shortcuts simultaneously. Notably, techniques such as DFR (Kirichenko et al., 2023) and GDRO (Sagawa et al., 2019) which are designed to reduce reliance on a specific shortcut feature, fail to make the model robust to unknown shortcuts effectively. In contrast, our experiments suggest that annotation-free methods can mitigate the impact of both labeled and unlabeled shortcut features more effectively.

Our evaluation of EVaLS is based on the spurious correlation benchmarks. This is because, in other instances of subpopulation shift, the attributes that differ across groups are not predictive of the label, thereby reducing the visibility of these attributes' effects in the model's final layers (Lee et al., 2023). Consequently, EIIL, which depends on output logits for prediction, might not effectively separate the groups. This observation is further supported by our findings related to the degree of group shift between the environments inferred by EIIL for each class in the CivilComments and MultiNLI datasets. The average group shift (defined in the Section 3.2) in the environments of the minority class of CivilComments is only $0.8_{\pm 0.0}\%$. Also, environments associated with Classes 1 and 2 in MultiNLI show only $1.1_{\pm 0.3}\%$ and $1.9_{\pm 1.0}\%$ group shift respectively. More results and ablation studies can be found in the Appendix.

**Mitigating Multiple Spurious Attributes** To evaluate the performance of our method in the case of unknown spurious correlations, we train a ResNet-18 He et al. (2016) model on the *Dominoes-CMF* dataset. We apply DFR Kirichenko et al. (2023), EVaLS-GL, and EVaLS on top of the trained ERMs to assess their ability to mitigate multiple shortcuts. We consider the style (MNIST/Fashion-MNIST) feature as the known group label, and the color as the unknown spurious attribute. We set the spurious correlation of the known attribute to $75\%$ and conduct experiments for various amount of unknown spurious correlation. During model selection, we calculate the worst-group accuracy on the validation set considering only the label of the known shortcut, *i.e.*, the lowest accuracy among the four groups based on the combination of the target label and the single known shortcut label. However, the final results on test data are based on the worst group accuracies, taking into account groups defined by the labels of both spurious attributes. The results are shown in Figure 4(b). Note that EVaLS operates without using annotations for either the known or unknown spurious attributes.

Our results confirm findings by Li et al. (2023), suggesting that methods using group labels mitigate reliance on the known shortcut but not necessarily on the unknown one. DFR (Kirichenko et al., 2023) experiences a significant drop in performance ($34.55\%$ under $95\%$ color spurious correlation) when it relies on a single known spurious attribute for grouping, compared to the oracle

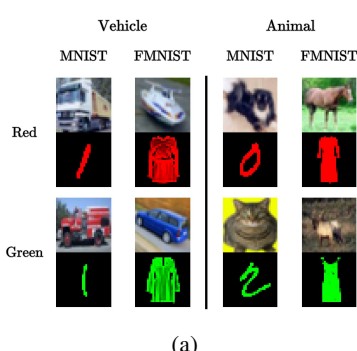 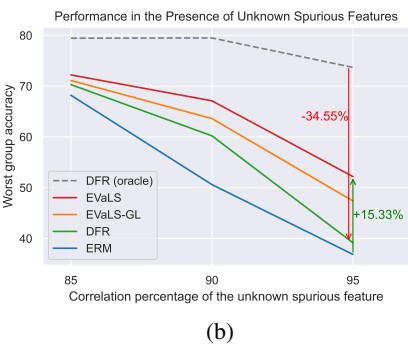

(a) (b)

Figure 4: (a) The Dominoes-CMF dataset, which contains two spurious attributes. (b) Performance on Dominoes-CMF is measured by worst-group accuracy across varying levels of correlation between the target label and the unknown spurious attribute (color). Lower reliance on available group annotations (based on known spurious attributes, i.e., style) results in higher robustness to both attributes. The performance gap between EVaLS and EVaLS-GL with lower group supervision compared to DFR (Kirichenko et al., 2023) increases with higher correlations. The oracle uses DFR (Kirichenko et al., 2023) with complete group information regarding both attributes.

that uses both attributes for grouping. EVaLS-GL reduces this issue using its loss-based sampling approach, but surprisingly EVaLS even outperforms EVaLS-GL. Combining a loss-based sampling approach for last layer training and environment-based model selection, results in a completely group-annotation-free method in a multi-shortcut setting with unknown spurious correlations, and successfully re-weights features to perform well with respect to multiple spurious attributes. It is also evident that increasing unknown spurious correlation results in a larger gap between the performance of EVaLS and EVaLS-GL compared to DFR (Kirichenko et al., 2023).

## 5 DISCUSSION

This study presents EVaLS, a novel approach to improve robustness to spurious correlations with zero group annotation. EVaLS uses loss-based sampling to create a balanced training dataset that effectively disrupts spurious correlations and employs EIIL to infer environments for model selection. We also explore situations with multiple spurious correlations, some of which are unknown. In this context, we introduce Dominoes-CMF, a dataset in which two factors are spuriously correlated with the label, but only one is identified. Our findings suggest that EVaLS attains near-optimal worst test group accuracy on spurious correlation datasets. We also present EVaLS-GL, which needs group labels only for model selection. Our empirical tests on various datasets demonstrate that EVaLS-GL outperforms state-of-the-art methods requiring group labels during evaluation or training.

Note that this paper remains consistent with the findings of Lin et al. (2022). Our approach does not involve identifying spurious attributes without auxiliary information. Instead, the objective is to make a trained model robust against its reliance on shortcuts. Specifically, conditioning on what a trained model learns, we ascertain that both the loss value and the model's feature space are instrumental in mitigating shortcuts.

EVaLS and EVaLS-GL may struggle with small datasets due to a low number of selected samples for the last layer training. Also, as environment inference from the last layer features is not effective for all types of subpopulation shifts, EVaLS is limited to datasets with spurious correlation. Similar to other methods in the field, EVaLS prioritizes the worst group accuracy at the cost of less average accuracy. Additionally, a notable variance has been observed in some of our experiments.

EVaLS represents a significant advancement in the development of methods for enhancing model fairness and robustness without prior knowledge about group annotations. EVaLS could be simply applied as a plug-and-play solution on various ERM-pretrained models with unknown inherent biases to make them robust to possible spurious correlations. Future work could explore developing environment inference methods effective for other types of subpopulation shift, such as attribute and class imbalance.

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

# A   RELATED WORK

Robustness to spurious correlation is a critical concern across various machine learning subfields. It is a form of out-of-distribution generalization (Shen et al., 2021) where the distribution shift arises from the disproportionate representation of minority groups—those instances that are devoid of the correlated spurious patterns associated with their labels (Yang et al., 2023b). The issue of spurious correlation also intersects with the discourse on fairness in machine learning (Seo et al., 2022; Mao et al., 2023).

Past studies have proposed a range of strategies to mitigate the models' reliance on spurious correlation. Broadly speaking, these methods can be categorized according to the degree of supervision they require regarding group labels.

Invariant learning (IL) methods  (Arjovsky et al., 2020; Krueger et al., 2021; Rame et al., 2022) operate under the assumption of having access to a collection of environments that comprise group shift. By imposing invariant conditions on these environments, IL methods strive to create classifiers robust against group-sensitive features. IRM (Arjovsky et al., 2020) is designed to learn a feature extractor, which, when utilized, guarantees the existence of a classifier that would be optimal in all training environments. VREx (Krueger et al., 2021) aims to decrease the risk variance among different training environments. PGI (Ahmed et al., 2021) works by minimizing the distance between the expected softmax distribution of labels, conditioned on inputs across both majority and minority environments. Lastly, Fishr (Rame et al., 2022) focuses on bringing the variance of risk gradients closer together across different training environments. For scenarios that the environments are not available, environment inference methods (Creager et al., 2021; Liu et al., 2021b) are used to obtain a set of environments. Creager et al. (2021) introduce environment inference for invariant learning (EIIL), which tries to partition samples into two groups such that the objective of IRM (Arjovsky et al., 2020) is maximized. HRM (Liu et al., 2021b) aims to optimize both an environment inference module and an invariant prediction module jointly, with the goal of achieving an invariant predictor.

When group annotations are accessible, various methods leverage this information to equalize the impact of different groups on the model's loss. The Group Distributionally Robust Optimization (GDRO) approach (Sagawa et al., 2019), for instance, focuses on optimizing the loss for the worst-performing group during training. Kirichenko et al. (2023) has shown that models can still learn and extract core data features even in the presence high spurious correlation. Consequently, They suggest that retraining just the last layer of a model initially trained with Empirical Risk Minimization (ERM) can effectively reduce reliance on spurious correlation for predicting class labels. This method, termed Deep Feature Re-weighting (DFR), has been validated as not only highly effective but also significantly more efficient than earlier techniques that necessitated retraining the full model (Nam et al., 2021; Sagawa et al., 2019). However, availability of group annotations is considered a serious restrictive assumption.

Several recent studies have endeavored to enhance model robustness against spurious correlation, even in the absence of group annotations (Liu et al., 2021a; Zhang et al., 2021; Qiu et al., 2023; LaBonte et al., 2023; Yang et al., 2023a). Liu et al. (2021a) introduce a two-stage method that involves training a model using ERM for a number of epochs before retraining it to give more weight to misclassified samples. The study by Zhang et al. (2021) employs the same two-stage training process, but with a twist for the second stage: they utilize contrastive methods. The goal is to bring samples from the same class but with divergent predictions closer in the feature space, while simultaneously increasing the separation between samples from different classes that have similar predictions. Another method, known as automatic feature reweighting (AFR) (Qiu et al., 2023), reweights the last layer of an ERM-pretrained model to favor samples that the original model was less accurate on. LaBonte et al. (2023) refine the last layer of an ERM-trained model through class-balanced finetuning, identifying challenging data points by comparing the classifier's predictions with those of an early-stopped version. While these methods have significantly reduced the reliance on group annotations, they still required for validation and model selection. This remains a constraint, particularly when the spurious correlation is completely unknown.

To make a trained model robust to subpopulation shifts with zero group annotations, LaBonte et al. (2023) have recently demonstrated that class-balanced retraining of a model pretrained with ERM can effectively improve the worst-group accuracy (WGA) for certain datasets. While this method effectively reduces the impact of class imbalance, it fails in datasets with spurious correlations.

Table 2: The average and variation percentage (%)(across 3 seeds) of group shift between the inferred environments using EIIL (Creager et al., 2021) for each class, which is the absolute difference between the proportion of a minority group in the two environments of a class. Higher group shift indicates better separation of environments. In most cases, a significant group shift is observed between the inferred environments.

| Class No. | Dataset | | |
| --- | --- | --- | --- |
| | Waterbirds | CelebA | UrbanCars |
| 0 | $16.6_{\pm 0.7}$ | $3.6_{\pm 0.2}$ | $17.7_{\pm 1.2}, 23.5_{\pm 0.1}, 62.1_{\pm 1.9}$ |
| 1 | $50.5_{\pm 0.3}$ | $14.1_{\pm 0.9}$ | $40.7_{\pm 7.9}, 13.8_{\pm 0.1}, 19.2_{\pm 3.9}$ |

## B  ENVIRONMENT INFERENCE FOR INVARIANT LEARNING

Consider the training dataset $\mathcal{D}^{\text{Tr}} = \{(x^{(i)}, y^{(i)})|x^{(i)} \in \mathcal{X}, y^{(i)} \in \mathcal{Y}\}$, where $\mathcal{X}$ and $\mathcal{Y}$ represent the input and output spaces, respectively. This dataset can be partitioned into different environments $\mathcal{E}^{tr} = \{e_1, ..., e_n\}$, such that for any $i \neq j$, the data distribution in $e_i$ and $e_j$ differs. The objective of invariant learning is to train a predictor that performs consistently across all environments in $\mathcal{E}^{tr}$. Under certain conditions, this predictor is also expected to perform well on $e^{tst}$, a test environment with a distribution distinct from the training data. Invariant Risk Minimization (IRM) (Arjovsky et al., 2020) approaches this problem by learning a feature extractor $\Phi(.)$ such that a classifier $\omega(.)$ exists, where $\omega \circ \Phi(.)$ performs consistently across all training environments. The practical implementation of the IRM objective is to minimize

$$\sum_{e \in \mathcal{E}^{tr}} R^e(\Phi) + \lambda ||\nabla_{\bar{\omega}} R^e(\bar{\omega} \circ \Phi)||^2, \tag{3}$$

where $\bar{\omega}$ is a constant scalar with a value of 1.0, $\lambda$ is a hyperparameter, and $R^e(f) = \mathbb{E}_{(x,y) \sim p_e}[l(f(x), y)]$ is referred to as the risk on environment $e$.

In real-world scenarios, training environments might not always be available. To address this, Environment Inference for Invariant Learning (EIIL) (Creager et al., 2021) partitions samples into two environments in a way that maximizes the objective in Eq 3.

During the training phase, the EIIL algorithm replaces the hard assignment of environments to samples with a soft assignment $\mathbf{q}_i(e) = p(e|(x^{(i)}, y^{(i)}))$, where $\mathbf{q}_i$ is learnable. Consequently, the relaxed version of the risk function is defined as $\tilde{R}^e(\Phi) = \frac{1}{N} \sum_i^N \mathbf{q}_i(e)[l(\Phi(x^{(i)}), y^{(i)})]$. Given a model $\Phi$ that has been trained with ERM on the dataset, EIIL optimizes

$$\mathbf{q}^* = \arg\max_{\mathbf{q}} ||\nabla_{\bar{\omega}} \tilde{R}^e(\bar{\omega} \circ \Phi)||. \tag{4}$$

As discussed in Creager et al. (2021), using a biased base model $\Phi$ could lead to environments exhibiting varying degrees of spurious correlation. During the inference phase, the soft assignment is converted to a hard assignment. The average group shift between the inferred environments using EIIL is illustrated in Table 2.

## C  ALGORITHM

---

**Algorithm 1** EVaLS

---

1: **Input:** Held-out dataset $\mathcal{D}^{\text{Val}}$, ERM-trained model $f_{\text{ERM}}$, maximum $k$ value $k_{\max}$
2: **Output:** Optimal number of samples $k^*$, best model $f^*$, best performance wea$^*$
3: $(\mathcal{D}^{\text{LL}}, \mathcal{D}^{\text{MS}}) \leftarrow \text{splitDataset}(\mathcal{D}^{\text{Val}})$             ▷ Split the held-out dataset
4: $\text{Envs}[y] \leftarrow \text{inferEnvs}(\mathcal{D}^{\text{MS}})[y] \quad \forall y \in \mathcal{Y}$         ▷ Infer environments from $\mathcal{D}^{\text{MS}}$
5: $\text{sortedSamples}[y] \leftarrow \text{sortByLoss}(f_{\text{ERM}}, \mathcal{D}^{\text{LL}}[y]) \quad \forall y \in \mathcal{Y}$    ▷ Sort $\mathcal{D}^{\text{LL}}$ samples by their loss
6: Initialize wea$^* \leftarrow 0$, $k^* \leftarrow 0$, $f^* \leftarrow \text{None}$
7: **for** $k = 1$ to $k_{\max}$ **do**
8:      $\text{highLossSamples}[y] \leftarrow \text{sortedSamples}[y][: k] \quad \forall y \in \mathcal{Y}$    ▷ Select top-$k$ high-loss samples
9:      $\text{lowLossSamples}[y] \leftarrow \text{sortedSamples}[y][-k :] \quad \forall y \in \mathcal{Y}$    ▷ Select top-$k$ low-loss samples
10:      $\mathcal{D}^{\text{Bal}} \leftarrow \{\text{highLossSamples}, \text{lowLossSamples}\}$           ▷ Combine samples
11:      $f \leftarrow \text{retrainLastLayer}(\mathcal{D}^{\text{Bal}})$        ▷ Retrain the last layer with combined samples
12:      $\text{wea} \leftarrow \text{evaluateWEA}(f, \text{Envs})$          ▷ Evaluate the retrained model
13:      **if** wea $>$ wea$^*$ **then**
14:          wea$^* \leftarrow$ wea, $f^* \leftarrow f$, $k^* \leftarrow k$         ▷ Record the best configuration
15:      **end if**
16: **end for**
17: **Return:** $k^*$, wea$^*$, $f^*$

---

## D  THEORETICAL ANALYSIS

In this section, we establish a more formal description of loss-based sampling for balanced dataset creation and then prove it. We thoroughly analyze the close relationship between the availability of the balanced dataset and the gap between spurious features of minority and majority groups.

### D.1  FEASIBILITY OF LOSS-BASED GROUP BALANCING

Consider a binary classification problem with a cross-entropy loss function. Let logits be denoted as $L$. Because loss is a monotonic function of logits, the tails of the distribution of loss across samples are equivalent to that of the logits in each class. We assume that in feature space (output of $g_\theta$) samples from the minority and majority of a class are derived from Gaussian distributions $\mathcal{N}(h_{\min}, \Sigma_{\min})$ and $\mathcal{N}(h_{\text{maj}}, \Sigma_{\text{maj}})$, respectively. Before diving into the group balance problem we initially show that the distribution of minority and majority samples in the logit space (output of $h_\phi$) are Gaussian too.

**Lemma D.1.** *[Gaussain Distribution of Logits] Considering a Gaussian distribution $Z \sim \mathcal{N}(h, \Sigma)$ in feature space and $W \in \mathbb{R}^d$, then the distribution of logits is as follows: $L = \langle W, Z \rangle \sim \mathcal{N}(Wh, \|W\|_\Sigma^2)$.*

*Proof.* Let $Z \sim \mathcal{N}(h, \Sigma)$.

Consider $L = \langle W, Z \rangle = W^T Z$, where $W \in \mathbb{R}^d$. L is a linear combination of jointly gaussian random variables which makes it an univariate gaussian random variable.

To find the distribution of $L$, we need to determine its mean and variance.

1. **Mean of $L$**

$$\mathbb{E}[L] = \mathbb{E}[\langle W, Z \rangle] = \mathbb{E}[W^T Z] = W^T \mathbb{E}[Z] = W^T h = \langle W, h \rangle.$$

Therefore, the mean of $L$ is $Wh$.

2. **Variance of $L$:**

The variance of $L$ can be computed using the properties of covariance. Recall that if $Z \sim \mathcal{N}(h, \Sigma)$, then the covariance matrix of $Z$ is $\Sigma$.

The variance of the linear combination $L = W^T Z$ is given by:

$$\text{Var}(L) = \text{Var}(W^T Z) = W^T \Sigma W = \|W\|_\Sigma^2,$$

where $\|W\|_\Sigma$ denotes the Mahalanobis norm of $W$.

Thus, we have proved that if $Z \sim \mathcal{N}(h, \Sigma)$, then the logits $L = \langle W, Z \rangle$ follow the distribution $\mathcal{N}(Wh, \|W\|_\Sigma^2)$. □

From now on, we consider $\mathcal{N}(\mu_{\min}, \sigma_{\min}^2)$ and $\mathcal{N}(\mu_{\text{maj}}, \sigma_{\text{maj}}^2)$ as the distribution of minority and majority samples in logits space.

Next, we prove the more formal version of the main proposition 3.1, which describes the existence of a balanced dataset, only after we define a key concept, *proportional density difference* (illustrated in figure 5) to outline our proof.

**Definition D.1** (Proportional Density Difference). *For any interval $I = (a, b]$ and a mixture distribution $\varepsilon P_1(x) + (1 - \varepsilon) P_2(x)$, proportional density difference is defined by the difference of accumulation of two component distributions in the interval $I$ and is denoted by $\Delta_\varepsilon P_{mixture}(I)$.*

$$\Delta_\varepsilon P_{mixture}(I) \overset{\Delta}{=} \varepsilon P_1\big(x \in I\big) - (1 - \varepsilon) P_2\big(x \in I\big)$$

**Definition D.2** (Tail Proportional Density Difference). *For a mixture distribution $\varepsilon P_1(x) + (1 - \varepsilon) P_2(x)$, we define $tail_L(\alpha)$ as $\Delta_\varepsilon P_{mixture}\big((-\infty, \alpha]\big)$ and $tail_R(\beta)$ as $-\Delta_\varepsilon P_{mixture}\big((\beta, +\infty)\big)$.*

**Corollary D.1.**

$$tail_L(\alpha) = \varepsilon F^1(\alpha) - (1 - \varepsilon) F^2(\alpha)$$

$$tail_R(\beta) = (1 - \varepsilon)\big[1 - F^2(\beta)\big] - \varepsilon\big[1 - F^1(\beta)\big]$$

*where $F^1$ and $F^2$ are CDF of two component distributions.*

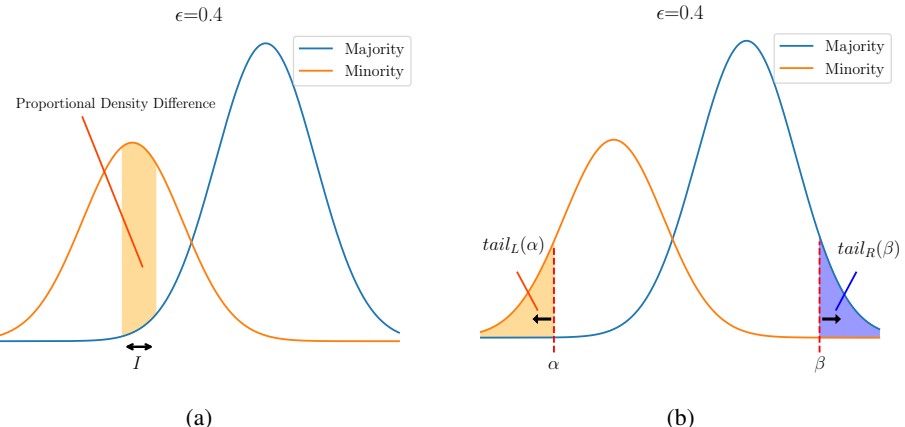

Figure 5: (a) Illustration of proportion density difference D.1, (b) equation of $tail_L(\alpha) = tail_R(\beta)$ at D.2.

**Proposition D.1.** *[Feasiblity Of Loss-based Group Balancing] Suppose that $L$ is derived from the mixture of two distributions $\mathcal{N}(\mu_{min}, \sigma_{min}^2)$ and $\mathcal{N}(\mu_{maj}, \sigma_{maj}^2)$ with proportion of $\varepsilon$ and $1 - \varepsilon$, respectively, where $\varepsilon \leq \frac{1}{2}$. There exists $\alpha$ and $\beta$ such that restricting $L$ to the $\alpha$-left and $\beta$-right tails of its distribution results in a group-balanced distribution if and only if (i)*

$$\sigma_{min} \geq \sigma_{maj}, \tag{5}$$

*or (ii)*

$$tail_L(\frac{-B + \sqrt{\Delta}}{2A}) > 0 \tag{6}$$

*and*

$$\epsilon \geq sigmoid\left( -\frac{(\mu_{maj} - \mu_{min})^2}{2(\sigma_{maj}^2 - \sigma_{min}^2)} - \log\left(\frac{\sigma_{maj}}{\sigma_{min}}\right) \right). \tag{7}$$

*where $A = \left(\frac{1}{2\sigma_{maj}^2} - \frac{1}{2\sigma_{min}^2}\right)$, $B = \left(\frac{\mu_{min}}{\sigma_{min}^2} - \frac{\mu_{maj}}{\sigma_{maj}^2}\right)$ and $\Delta = \frac{(\mu_{min} - \mu_{maj})^2}{\sigma_{min}^2 \sigma_{maj}^2} - 4\left[\log\left(\frac{\sigma_{maj}}{\sigma_{min}}\right) + \log\left(\frac{\epsilon}{1-\epsilon}\right)\right]\left[\frac{1}{2\sigma_{maj}^2} - \frac{1}{2\sigma_{min}^2}\right]$.*

**Proof outline**

Our proof proceeds with three steps. First, we reformulate the theorem as an equality of left- and right-tail proportional distribution differences. In other words, we show that the more mass the minority distribution has on one tail, the more mass the majority distribution must have on the other tail. Afterward, supposing $\mu_{\min} < \mu_{\text{maj}}$ WLOG, we propose a proper range for $\beta$ values on the right tail. We show that when $\sigma_{\text{maj}} \leq \sigma_{\min}$, values for $\alpha$ trivially exist that can overcome the imbalance between the two distributions. In the last step, for the case in which the variance of the majority is higher than the minority, we discuss a necessary and sufficient condition for the existence of $\alpha$ and $\beta$ based on the left-tail proportional density difference using the properties of its derivative with respect to $\alpha$.

**Step 1** *Reformulating the problem based on proportional distribution difference.*

We introduce a utility random variable *Logit Value Tier* as $T$, which is defined as a function of a random variable $L$.

$$T_{\alpha,\beta} = \begin{cases} High & \text{if } L \geq \beta \\ Mid & \text{if } \alpha < L < \beta \\ Low & \text{if } L \leq \alpha \end{cases} \tag{8}$$

We can rewrite the problem in formal form as finding an $\alpha$ and $\beta$ which satisfies the following equation:

$$P\Big(g = \text{min}\Big|T_{\alpha,\beta} \neq Mid\Big) = P\Big(g = \text{maj}\Big|T_{\alpha,\beta} \neq Mid\Big) \tag{9}$$

Equation 7 now can be rewritten to a more suitable form:

$$P\Big(g = \mathrm{min}\Big|T_{\alpha,\beta} \neq Mid\Big) = P\Big(g = \mathrm{maj}\Big|T_{\alpha,\beta} \neq Mid\Big) \tag{10}$$

$$\Longleftrightarrow \quad \frac{P\Big(T_{\alpha,\beta} \neq Mid\Big|g = \mathrm{min}\Big)P\Big(g = \mathrm{min}\Big)}{P\Big(T_{\alpha,\beta} \neq Mid\Big)} = \frac{P\Big(T_{\alpha,\beta} \neq Mid|g = \mathrm{maj}\Big)P\Big(g = \mathrm{maj}\Big)}{P\Big(T_{\alpha,\beta} \neq Mid\Big)} \tag{11}$$

$$\Longleftrightarrow \quad P\Big(T_{\alpha,\beta} \neq Mid\Big|g = \mathrm{min}\Big)P\Big(g = \mathrm{min}\Big) = P\Big(T_{\alpha,\beta} \neq Mid\Big|g = \mathrm{maj}\Big)P\Big(g = \mathrm{maj}\Big) \tag{12}$$

$$\Longleftrightarrow \quad \varepsilon P\Big(T_{\alpha,\beta} \neq Mid\Big|g = \mathrm{min}\Big) = (1-\varepsilon)P\Big(T_{\alpha,\beta} \neq Mid\Big|g = \mathrm{maj}\Big) \tag{13}$$

$$\Longleftrightarrow \quad \varepsilon\bigg[P\Big(T_{\alpha,\beta} = Low\Big|g = \mathrm{min}\Big) + P\Big(T_{\alpha,\beta} = High\Big|g = \mathrm{min}\Big)\bigg] = \tag{14}$$

$$(1-\varepsilon)\bigg[P\Big(T_{\alpha,\beta} = Low\Big|g = \mathrm{maj}\Big) + P\Big(T_{\alpha,\beta} = High\Big|g = \mathrm{maj}\Big)\bigg] \tag{15}$$

$$\Longleftrightarrow \quad \varepsilon\bigg[P\Big(L \leq \alpha\Big|g = \mathrm{min}\Big) + P\Big(L \geq \beta\Big|g = \mathrm{min}\Big)\bigg] = \tag{16}$$

$$(1-\varepsilon)\bigg[P\Big(L \leq \alpha\Big|g = \mathrm{maj}\Big) + P\Big(L \geq \beta\Big|g = \mathrm{maj}\Big)\bigg] \tag{17}$$

$$\Longleftrightarrow \quad \varepsilon\bigg[F^{\mathrm{min}}(\alpha) + \Big(1 - F^{\mathrm{min}}(\beta)\Big)\bigg] = (1-\varepsilon)\bigg[F^{\mathrm{maj}}(\alpha) + \Big(1 - F^{\mathrm{maj}}(\beta)\Big)\bigg] \tag{18}$$

$$\Longleftrightarrow \quad \varepsilon F^{\mathrm{min}}(\alpha) - (1-\varepsilon)F^{\mathrm{maj}}(\alpha) = (1-\varepsilon)\Big[1 - F^{\mathrm{maj}}(\beta)\Big] - \varepsilon\Big[1 - F^{\mathrm{min}}(\beta)\Big] \tag{19}$$

We can see the left side of equation 19 is just a function of $alpha$. The same goes for the right side of the equation which is a function of $\beta$.

Rewriting the left side of the equation as $tail_L(\alpha)$ and right side as $tail_R(\beta)$, the problem is now reduced to finding an $\alpha$ and $\beta$ that satisfies

$$tail_L(\alpha) = tail_R(\beta) \tag{20}$$

which is shown in figure 5.

Before reaching out to step two we discuss the properties of $tail_L$ and $tail_R$ in Lemma D.2.

**Lemma D.2.** *$tail_L(\alpha)$ and $tail_R(\beta)$ are continuous functions and $\lim_{\alpha \to -\infty} tail_L(\alpha) = 0$, $\lim_{\alpha \to +\infty} tail_L(\alpha) = 2\varepsilon - 1 < 0$, $\lim_{\beta \to +\infty} tail_R(\beta) = 0$ and $\lim_{\beta \to -\infty} tail_R(\beta) = 1 - 2\varepsilon > 0$.*

*Proof.* Simply proved by the definition of $tail$ functions and properties of CDF. $\qquad\square$

**Step 2** *Solving the equation 20 for simple cases.*

**Lemma D.3.** $tail_R(\mu_{maj}) > \frac{1}{2} - \varepsilon \geq 0$

*Proof.*

$$tail_R(\mu_{\mathrm{maj}}) = (1-\varepsilon)\Big[1 - F^{\mathrm{maj}}(\mu_{\mathrm{maj}})\Big] - \varepsilon\Big[1 - F^{\mathrm{min}}(\mu_{\mathrm{maj}})\Big] \tag{21}$$

$$= (1-\varepsilon)\Big[1 - \phi(0)\Big] - \varepsilon\Big[1 - \phi\big(\frac{\mu_{\mathrm{maj}} - \mu_{\mathrm{min}}}{\sigma_{\mathrm{min}}}\big)\Big] \tag{22}$$

$$> \frac{(1-\varepsilon)}{2} - \varepsilon\big(1 - \frac{1}{2}\big) = \frac{1 - 2\varepsilon}{2} = \frac{1}{2} - \varepsilon \tag{23}$$

$\square$

**Corollary D.2.** *Because $tail_R$ is continuous and $\lim_{\beta \to +\infty} tail_R(\beta) = 0$, based on the mean value theorem, any value between zero and $\frac{(1-2\varepsilon)}{2}$ is obtainable by selecting a $\beta$ in $[\mu_2, +\infty)$.*

According to the previous corollary D.2 finding a positive $tail_L(\alpha)$ will satisfy our need. to find a suitable point, we employ derivatives and properties of relative PDFs to maximize $tail_L(\alpha)$ and find a positive value.

$$\frac{\mathrm{d}tail_L(\alpha)}{\mathrm{d}\alpha} = \varepsilon f^{\min}(\alpha) - (1-\varepsilon)f^{\text{maj}}(\alpha) = \varepsilon f^{\text{maj}}(\alpha)\left[\frac{f^{\min}(\alpha)}{f^{\text{maj}}(\alpha)} - \frac{1-\varepsilon}{\varepsilon}\right] \quad (24)$$

The term $\left[\frac{f^{\min}(\alpha)}{f^{\text{maj}}(\alpha)} - \frac{1-\varepsilon}{\varepsilon}\right]$ has the same sign with derivative of $tail_L(\alpha)$, also it's roots are critical points of $tail_L$, analyzing characteristics of $\log \frac{f^{\min}(\alpha)}{f^{\text{maj}}(\alpha)}$ is the key insight to find a proper $\alpha$ value.

$$\log f^{\min}(\alpha) - \log f^{\text{maj}}(\alpha) = \log\left(\frac{1-\epsilon}{\epsilon}\right)$$

$$\Rightarrow \log\left(\frac{\sigma_{\text{maj}}}{\sigma_{\min}}\right) - \log\left(\frac{1-\epsilon}{\epsilon}\right) - \frac{(\alpha - \mu_{\min})^2}{2\sigma_{\min}^2} + \frac{(\alpha - \mu_{\text{maj}})^2}{2\sigma_{\text{maj}}^2} = 0$$

$$\Rightarrow \left(\frac{1}{2\sigma_{\text{maj}}^2} - \frac{1}{2\sigma_{\min}^2}\right)\alpha^2 + \left(\frac{\mu_{\min}}{\sigma_{\min}^2} - \frac{\mu_{\text{maj}}}{\sigma_{\text{maj}}^2}\right)\alpha + \left[\frac{\mu_{\text{maj}}^2}{2\sigma_{\text{maj}}^2} - \frac{\mu_{\min}^2}{2\sigma_{\min}^2} + \log\left(\frac{\sigma_{\text{maj}}}{\sigma_{\min}}\right) + \log\left(\frac{\epsilon}{1-\epsilon}\right)\right] = 0$$

Because $\lim_{\alpha \to -\infty} tail_L(\alpha) = 0$ and $\lim_{\beta \to +\infty} tail_R(\beta) < 0$ to have a positive $tail_L(\alpha)$, we need to have an interval which $\frac{\mathrm{d}tail_L(\alpha)}{\mathrm{d}\alpha}$ is positive. For a second degree polynomial like $ax^2 + bx + c$ to have positive value, either $a \geq 0$ or $\Delta > 0$, in our case $a$ is $\left(\frac{1}{\sigma_{\text{maj}}^2} - \frac{1}{\sigma_{\min}^2}\right)$. if $\sigma_{\min} \geq \sigma_{\text{maj}}$ then $a \geq 0$ and the minority CDF function will dominate the majority CDF function in the left-side tail and by choosing a negative number with big enough absolute value for alpha and $tail_L(\alpha)$ will be positive.

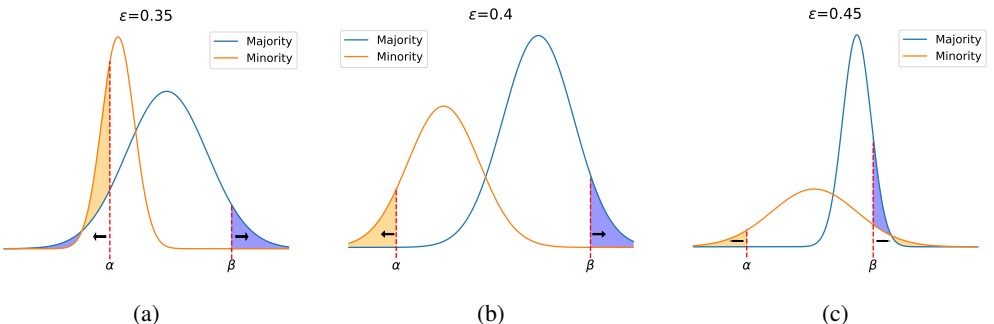

Figure 6: Tail thresholds for three cases: (a) minority group variance is less than majority ($\sigma_{\min} < \sigma_{\text{maj}}$), (b) the variance of two groups are equal ($\sigma_{\min} = \sigma_{\text{maj}}$) and (c) the variance of the minority group is more than majority ($\sigma_{\min} > \sigma_{\text{maj}}$).

**Step 3** *Solving equation 20 for special case $\sigma_{min} < \sigma_{maj}$* In case of $\sigma_{\min} \leq \sigma_{\text{maj}}$, having $\Delta > 0$ is a necessary condition, also derivative of $tail_L(\alpha)$ is only positive in $\left(\frac{-b-\sqrt{\Delta}}{2a}, \frac{-b+\sqrt{\Delta}}{2a}\right)$ so the maximum of $tail_L$ is either in $-\infty$ or in $\frac{-b+\sqrt{\Delta}}{2a}$. Having $tail_L\left(\frac{-b+\sqrt{\Delta}}{2a}\right) > 0$ next to $\Delta > 0$ condition, would be the necessary and also sufficient in this case.

$$B^2 = \frac{\mu_{\min}^2}{\sigma_{\min}^4} + \frac{\mu_{\text{maj}}^2}{\sigma_{\text{maj}}^4} - 2\frac{\mu_{\text{maj}}\mu_{\min}}{\sigma_{\text{maj}}^2 \sigma_{\min}^2}$$

$$4AC = \frac{\mu_{\min}^2}{\sigma_{\min}^4} - \frac{\mu_{\min}^2}{\sigma_{\maj}^2\sigma_{\min}^2} - \frac{\mu_{\maj}^2}{\sigma_{\maj}^2\sigma_{\min}^2} + \frac{\mu_{\maj}^2}{\sigma_{\maj}^4} + 4\left[\log\left(\frac{\sigma_{\maj}}{\sigma_{\min}}\right) + \log\left(\frac{\epsilon}{1-\epsilon}\right)\right]\left[\frac{1}{2\sigma_{\maj}^2} - \frac{1}{2\sigma_{\min}^2}\right]$$

$$\Delta = \frac{(\mu_{\min}-\mu_{\maj})^2}{\sigma_{\min}^2\sigma_{\maj}^2} - 4\left[\log\left(\frac{\sigma_{\maj}}{\sigma_{\min}}\right) + \log\left(\frac{\epsilon}{1-\epsilon}\right)\right]\left[\frac{1}{2\sigma_{\maj}^2} - \frac{1}{2\sigma_{\min}^2}\right] \geq 0$$

$$\iff (\mu_{\min}-\mu_{\maj})^2 \geq 2\left[\log\left(\frac{1-\epsilon}{\epsilon}\right) - \log\left(\frac{\sigma_{\maj}}{\sigma_{\min}}\right)\right]\left[\sigma_{\maj}^2 - \sigma_{\min}^2\right]$$

$$\iff \epsilon \geq \text{sigmoid}\left(-\frac{(\mu_{\maj}-\mu_{\min})^2}{2(\sigma_{\maj}^2-\sigma_{\min}^2)} - \log\left(\frac{\sigma_{\maj}}{\sigma_{\min}}\right)\right)$$

Next, we investigate properties of the conditions of the proposition D.1 in case of $\sigma_{\maj} < \sigma_{\min}$. Schematic interpretation of these conditions is presented in figure 7.

- As equation 7 indicates, the minority group is not allowed to be too underrepresented. This especially has a direct relation with the difference of means. The more mean values of groups are different, the more imbalance can be mitigated through loss-based sampling. Mean value difference is especially affected by the spurious correlation, it escalates as the model relies on spurious correlation and also when the spurious features between groups are too different.
- On the other hand condition 6 is more complex and doesn't have a simple closed form, we analytically describe its behaviors by fixating the means and calculating the valid values for $\varepsilon$. As the results show in figure 7, most of $\varepsilon$ are feasible in for $\sigma_{\min} < \Delta\mu$ as we can see the possible region declines with an increase of $\sigma_{\min}$ and valid $\varepsilon$ values cease to exist.

### D.2 PRACTICAL JUSTIFICATION

As shown in Table 3, the standard deviation ($\sigma$) of the minority group is consistently greater than that of the majority group across all analyzed datasets. Consequently, condition $(i)$ (Eq. 5) of Proposition D.1 is satisfied. Therefore, we theoretically expect the existence of properly balanced left and right tails.

Table 3: Means, standard deviations (STD), and Earth Mover's Distance across WaterBirds and CelebA datasets.

| | Waterbirds | | | | CelebA | |
| | Class 1 | | Class 2 | | Class 2 | |
| | Min | Maj | Min | Maj | Min | Maj |
|---|---|---|---|---|---|---|
| **Mean ($\mu$)** | $-6.77$ | $-19.17$ | $2.55$ | $11.39$ | $-1.02$ | $6.42$ |
| **STD ($\sigma$)** | $6.31$ | $6.23$ | $6.97$ | $4.75$ | $7.64$ | $6.48$ |
| **Earth Mover's Distance** | $12.40$ | | $8.84$ | | $7.43$ | |

## E EXPERIMENTAL DETAILS

### E.1 COMPLETE RESULTS

The complete results on Waterbirds, CelebA, and UrbanCars, in addition to complete results on CivilComments and MultiNLI are reported in Tables 4 and 5 respectively. The results for all methods except Group DRO + EIIL on all datasets except UrbanCars are reported by Qiu et al. (2023). The results for Group DRO + EIIL are taken from Zhang et al. (2021). Also, the results of our method and DFR are shown in Table 6

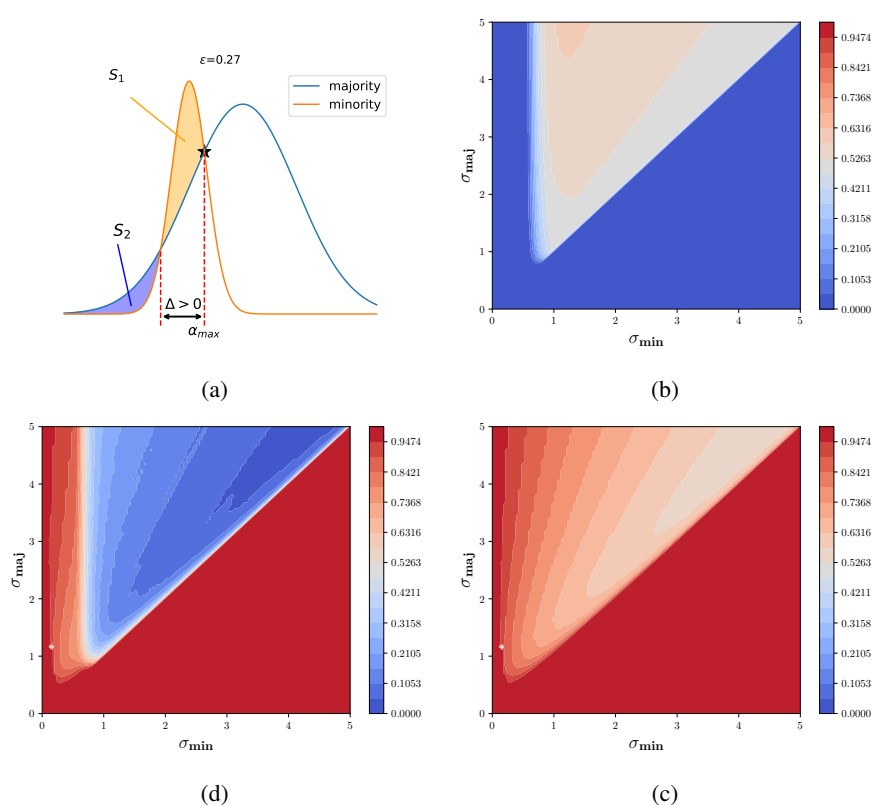

Figure 7: (a) Conditions if $\sigma_{\min} > \sigma_{\text{maj}}$, (b), (c), (d) minimum, maximum and interval length of feasible $\varepsilon$ values across $(\sigma_{\min}, \sigma_{\text{maj}})$ field for $\mu_{\min} = 0$, $\mu_{\text{maj}} = 1$.

Table 4: A comparison of the various methods, ours included, on spurious correlation datasets. The Group Info column indicates if each method utilizes group labels of the training/validation data, with ✔ denoting that group information is employed during both the training and validation stages. Both the average test accuracy and worst test group accuracy are reported. The mean and standard deviation are calculated over three runs with different seeds. The numbers in bold represent the highest results among all methods, while the underlined numbers represent the best results among methods that may not require group annotation in the training phase.

| Method | Group Info Train/Val | Waterbirds Worst | Waterbirds Average | CelebA Worst | CelebA Average | UrbanCars Worst | UrbanCars Average |
|---|---|---|---|---|---|---|---|
| GDRO (Sagawa et al., 2019) | ✔/✔ | 91.4 | 93.5 | **88.9** | 92.9 | 73.1 | $84.2_{\pm1.3}$ |
| DFR (Kirichenko et al., 2023) | ✗/✔ | $\mathbf{92.9_{\pm0.2}}$ | $94.2_{\pm0.4}$ | $88.3_{\pm1.1}$ | $91.3_{\pm0.3}$ | $79.6_{\pm2.22}$ | $87.5_{\pm0.6}$ |
| GDRO + EIIL (Creager et al., 2021) | ✗/✔ | $77.2_{\pm1}$ | $\mathbf{96.5_{\pm0.2}}$ | $81.7_{\pm0.8}$ | $85.7_{\pm0.1}$ | $76.5_{\pm2.6}$ | $85.4_{\pm2.1}$ |
| JTT (Liu et al., 2021a) | ✗/✔ | 86.7 | $\underline{93.3}$ | 81.1 | 88.0 | 79.5 | 86.3 |
| SELF (LaBonte et al., 2023) | ✗/✔ | $\underline{91.6_{\pm1.4}}$ | $93.6_{\pm1.1}$ | $83.9_{\pm0.9}$ | $91.7_{\pm0.4}$ | $83.2_{\pm0.8}$ | $\mathbf{90.0_{\pm0.5}}$ |
| AFR (Qiu et al., 2023) | ✗/✔ | $\underline{90.4_{\pm1.1}}$ | $94.2_{1.2}$ | $82.0_{\pm0.5}$ | $91.3_{\pm0.3}$ | $80.2_{\pm2.0}$ | $\underline{87.1_{\pm1.2}}$ |
| EVaLS-GL (Ours) | ✗/✔ | $89.4_{\pm0.3}$ | $95.1_{\pm0.3}$ | $84.6_{\pm1.6}$ | $91.1_{\pm0.6}$ | $\underline{83.5_{\pm1.7}}$ | $88.3_{\pm0.9}$ |
| ERM | ✗/✗ | $66.4_{\pm2.3}$ | $90.3_{\pm0.5}$ | $47.4_{\pm2.3}$ | $\mathbf{95.5_{\pm0.0}}$ | $18.67_{\pm2.01}$ | $76.5_{\pm4.6}$ |
| EVaLS (Ours) | ✗/✗ | $88.4_{\pm3.1}$ | $94.1_{\pm0.1}$ | $\underline{85.3_{\pm0.4}}$ | $\underline{89.4_{\pm0.5}}$ | $82.1_{\pm0.9}$ | $88.1_{\pm0.9}$ |

Table 5: A comparison of the various methods, ours included, on CivilComments and MultiNLI. The Group Info column indicates if each method utilizes group labels of the training/validation data, with $\checkmark\!\!\checkmark$ denoting that group information is employed during both the training and validation stages. Both the average test accuracy and worst test group accuracy are reported. The mean and standard deviation are calculated over three runs with different seeds. The numbers in bold represent the highest results among all methods, while the underlined numbers represent the best results among methods that may not require group annotation in the training phase.

| Method | Group Info | CivilComments | | MultiNLI | |
| --- | --- | --- | --- | --- | --- |
| | Train/Val | Worst | Average | Worst | Average |
| GDRO (Sagawa et al., 2019) | $\checkmark$/$\checkmark$ | **69.9** | 88.9 | **77.7** | 81.4 |
| DFR (Kirichenko et al., 2023) | $\times$/$\checkmark\!\!\checkmark$ | $70.1_{\pm 0.8}$ | $87.2_{\pm 0.3}$ | $74.7_{\pm 0.7}$ | $82.1_{\pm 0.2}$ |
| GDRO + EIIL (Creager et al., 2021) | $\times$/$\checkmark$ | $67.0_{\pm 2.4}$ | $90.5_{\pm 0.2}$ | $61.2_{\pm 0.5}$ | $79.4_{\pm 0.2}$ |
| JTT (Liu et al., 2021a) | $\times$/$\checkmark$ | $\underline{69.3}$ | 91.1 | 72.6 | 78.6 |
| SELF (LaBonte et al., 2023) | $\times$/$\checkmark$ | $65.9_{\pm 1.7}$ | $89.7_{\pm 0.6}$ | $70.7_{\pm 2.5}$ | $81.2_{\pm 0.7}$ |
| AFR (Qiu et al., 2023) | $\times$/$\checkmark$ | $68.7_{\pm 0.6}$ | $89.8_{\pm 0.6}$ | $73.4_{\pm 0.6}$ | $81.4_{\pm 0.2}$ |
| EVaLS-GL (Ours) | $\times$/$\checkmark$ | $68.0_{\pm 0.5}$ | $89.2_{\pm 0.3}$ | $\underline{75.1}_{\pm 1.2}$ | $81.6_{\pm 0.2}$ |
| ERM | $\times$/$\times$ | $61.2_{\pm 3.6}$ | $\mathbf{92.0}_{\pm 0.0}$ | $64.8_{\pm 1.9}$ | $\mathbf{82.6}_{\pm 0.0}$ |

Table 6: A Comparison of ERM, DFR, EVaLS, and EVaLS-GL on the Dominoes-CMF with different spurious correlations for the unknown feature. Both the worst and average of test group accuracies are presented. The mean and standard deviation are calculated based on runs with three distinct seeds.

| Method | 85% Corr. | | 90% Corr. | | 95% Corr. | |
| --- | --- | --- | --- | --- | --- | --- |
| | Worst | Average | Worst | Average | Worst | Average |
| ERM | $68.3_{\pm 1.5}$ | $97.1_{\pm 0.5}$ | $50.6_{\pm 1.0}$ | $96.1_{\pm 0.0}$ | $36.8_{\pm 2.0}$ | $95.4_{\pm 1.0}$ |
| DFR | $70.7_{\pm 0.5}$ | $86.2_{\pm 0.6}$ | $60.2_{\pm 1.2}$ | $84.6_{\pm 0.4}$ | $42.7_{\pm 2.7}$ | $81.5_{\pm 1.2}$ |
| AFR | $65.7_{\pm 0.2}$ | $94.2_{\pm 0.8}$ | $54.2_{\pm 0.2}$ | $94.9_{\pm 2.1}$ | $40.3_{\pm 0.5}$ | $95.9_{\pm 1.2}$ |
| AFR + EIIL | $69.1_{\pm 0.1}$ | $92_{\pm 1.3}$ | $61.5_{\pm 0.2}$ | $92.1_{\pm 1.9}$ | $40.4_{\pm 0.1}$ | $92.9_{\pm 1.5}$ |
| EVaLS-GL | $70.1_{\pm 2.9}$ | $82.5_{\pm 1.8}$ | $63.6_{\pm 1.3}$ | $78.7_{\pm 1.5}$ | $48.5_{\pm 0.8}$ | $77.0_{\pm 2.0}$ |
| EVaLS | $\mathbf{73.0}_{\pm 4.8}$ | $81.5_{\pm 1.8}$ | $\mathbf{67.1}_{\pm 4.2}$ | $78.6_{\pm 2.0}$ | $\mathbf{51.2}_{\pm 1.4}$ | $77.5_{\pm 2.5}$ |

### E.2 DOMINOES-COLORED-MNIST-FASHIONMNIST

**Dominoes-Colored-MNIST-FashionMNIST (Dominoes-CMF)** is a synthetic dataset. We adopt a similar approach to previous works Pagliardini et al. (2022b); Shah et al. (2020); Kirichenko et al. (2023) using a modified version of the *Dominoes* binary classification dataset. This dataset consists of images with the top half showing CIFAR-10 images Krizhevsky & Hinton (2009), divided into two meaningful classes: vehicles (airplane, car, ship, truck) and animals (cat, dog, horse, deer). The bottom half displays either MNIST Deng (2012) images from classes $\{0 - 3\}$ or Fashion-MNIST Xiao et al. (2017) images from classes $\{\text{T-shirt}, \text{Dress}, \text{Coat}, \text{Shirt}\}$. The complex feature (top half) serves as the core feature and the simple feature (bottom half) is linearly separable and correlated with the class label at 75%. Furthermore, inspired by the approaches in Zhang et al. (2021); Arjovsky et al. (2020), we intentionally introduce an additional spurious attribute by artificially coloring a subset of images as follows: for three different datasets, 85%, 90%, and 95% of the images in the bottom half of class $c_1$ are randomly assigned a red color in each respective dataset, while 15%, 10%, and 5% of the images are assigned a green color, respectively. The same procedure is applied inversely for class $c_2$.

See Table 7 for more details about the dataset statistics.

### E.3 DATASETS

**Waterbirds (Sagawa et al., 2019)** The dataset comprises images of diverse bird species, classified into two categories: waterbirds and landbirds. Each image features a bird set against a backdrop of

Table 7: *Dominoes-CMF* Dataset Statistics for 85%, 90%, and 95% Correlation

| Top part | | Bottom Part (85% Corr.) | | Bottom Part (90% Corr.) | | Bottom Part (95% Corr.) | |
|---|---|---|---|---|---|---|---|
| CIFAR-10 Class | Color | MNIST | Fashion-MNIST | MNIST | Fashion-MNIST | MNIST | Fashion-MNIST |
| $c_1$ (Vehicle) | Red | 12,750 | 4,250 | 13,500 | 4,500 | 14,250 | 4,750 |
| | Green | 2,250 | 750 | 1,500 | 500 | 750 | 250 |
| $c_2$ (Animal) | Red | 750 | 2,250 | 500 | 1,500 | 250 | 750 |
| | Green | 4,250 | 12,750 | 4,500 | 13,500 | 4,750 | 14,250 |
| **Total** | | 40,000 | | 40,000 | | 40,000 | |

Table 8: ERM Accuracies on *Dominoes-CMF* Dataset. The mean and standard deviation are reported based on three runs with different seeds.

| Top part | | Bottom Part (85% Corr.) | | Bottom Part (90% Corr.) | | Bottom Part (95% Corr.) | |
|---|---|---|---|---|---|---|---|
| CIFAR-10 Class | Color | MNIST | Fashion-MNIST | MNIST | Fashion-MNIST | MNIST | Fashion-MNIST |
| $c_1$ (Vehicle) | Red | $98.53_{\pm0.01}\%$ | $95.61_{\pm1.1}\%$ | $99.2_{\pm0.01}\%$ | $95.2_{\pm1.1}\%$ | $99.63_{\pm0.01}\%$ | $98.11_{\pm1.1}\%$ |
| | Green | $89.33_{\pm2.4}\%$ | $68.57_{\pm0.5}\%$ | $84.5_{\pm2.4}\%$ | $54.7_{\pm0.5}\%$ | $63.1_{\pm1.4}\%$ | $36.84_{\pm0.5}\%$ |
| $c_2$ (Animal) | Red | $68.28_{\pm2.6}\%$ | $86.18_{\pm2.4}\%$ | $56.8_{\pm5.6}\%$ | $86.7_{\pm2.4}\%$ | $39.13_{\pm1.6}\%$ | $68.53_{\pm2.4}\%$ |
| | Green | $93.97_{\pm0.5}\%$ | $98.36_{\pm0.2}\%$ | $96.2_{\pm0.5}\%$ | $99.3_{\pm0.2}\%$ | $97.92_{\pm0.5}\%$ | $99.25_{\pm0.2}\%$ |

either water or land. Interestingly, the background scene acts as a spurious feature in this classification task. Waterbirds are primarily shown against water backgrounds, and landbirds against land backgrounds. Consequently, waterbirds on water and landbirds on land form the minority groups in the training data. It's important to note that the validation dataset for waterbirds is group-balanced, meaning birds from each class are equally represented against both water and land backgrounds. This dataset is mainly categorized as a spurious correlation dataset.

**CelebA (Liu et al., 2014)** is a widely used dataset in image classification tasks, featuring annotations for 40 binary facial attributes such as hair color, gender, and age. Hair color classification is particularly prominent in literature focusing on spurious correlation robustness. Notably, gender serves as a spurious attribute within this dataset, where a significant majority 94% of individuals with blond hair are women, while men with blond hair represent a minority group. In addition to spurious correlation in the class of blond hair, this dataset also exhibits class imbalance.

**MultiNLI (Williams et al., 2017)** dataset involves a text classification task focused on determining the relationship between pairs of sentences: contradiction, entailment, or neutral. Sentences containing negation words such as "no" or "never" are under-represented in all three classes, inducing attribute imbalance in the dataset. Figure 8 illustrates the distinct behavior of this dataset compared to other datasets that contain spurious attributes.

**CivilComments (Borkan et al., 2019)** dataset, as part of the WILDS benchmark, involves a text classification task focused on labeling online comments as either "toxic" or "not toxic". Each comment is associated with 8 attributes, including gender (male, female), sexual orientation (LGBTQ), race (black, white), and religion (Christian, Muslim, or other), based on whether these characteristics are mentioned in the comment. While there is a small attribute imbalance in the dataset, it can categorized into datasets with class imbalance. The detailed proportion of each attribute in each class is described in Table 9. In this paper, we use the implementation of the dataset by the `WILDS` package (Koh et al., 2021).

Table 9: Proportion of attributes in each class for CivilComments dataset.

| Toxicity (Class) | Male | Female | LGBTQ | Christian | Muslim | Other Religions | Black | White |
|---|---|---|---|---|---|---|---|---|
| 0 | 0.11 | 0.12 | 0.03 | 0.10 | 0.05 | 0.02 | 0.03 | 0.05 |
| 1 | 0.14 | 0.15 | 0.08 | 0.08 | 0.10 | 0.03 | 0.1 | 0.14 |

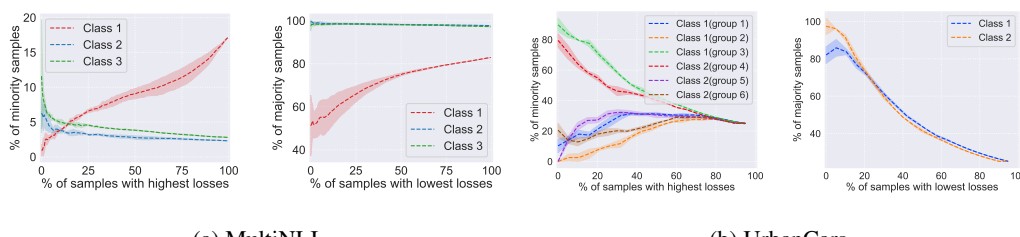

(a) MultiNLI          (b) UrbanCars

Figure 8: The percentage of samples with the highest (lowest) losses across various thresholds that belong to the minority (majority) group within different classes in $\mathcal{D}^{\text{LL}}$ for (a) MultiNLI and (b) UrbanCars datasets.

**UrbanCars (Li et al., 2023)** is an image classification dataset with multiple shortcuts. Each image in the dataset consists of a car in the center of the image on a natural scene background, with another object to the right of the image. Images are labeled *Urban* or *City* according to the type of car present in the center. However, each of the backgrounds and the additional objects is highly correlated with the label. While the test set consists of 8 environments based on combinations of the core and two spurious patterns, the training and validation set consist of four groups, based on combinations of the label and only one of the shortcuts.

### E.4 TRAINING DETAILS

**ERM** For Waterbirds and CelebA, we utilize the ResNet50 checkpoints available in the GitHub repository of Kirichenko et al. (2023) as our base model. We use the ResNet-50 architecture provided by the `torchvision` package. In the case of CivilComments and MultiNLI, we adopt a similar approach to Kirichenko et al. (2023), using `BertForSequenceClassification.from_pretrained ('bert-base-uncased', ...)` from the `transformers` package. The model is trained using the AdamW optimizer with a learning rate of $10^{-5}$, weight decay of $10^{-4}$, and a batch size of 16 for a total of 5 epochs.

For the UrbanCars dataset, we adhere to the settings described in Li et al. (2023), which involves training a ResNet-50 model pretrained on ImageNet using the SGD optimizer with a learning rate of $10^{-3}$, momentum of 0.9, weight decay of $10^{-4}$, and a batch size of 128 for 300 epochs. For the Dominoes-CMF dataset, we train a ResNet18 model pretrained on ImageNet for 20 epochs with a batch size of 128 and an SGD optimizer with a learning rate of $10^{-3}$, momentum of 0.9, and weight decay of $10^{-4}$.

**EVaLS and EVaLS-GL** For every dataset, EIIL was utilized with a learning rate of $0.01$, a total of 20000 steps, and a batch size of 128. The last layer of the model was trained on all datasets using the Adam optimizer. A batch size of 32 and a weight decay of $10^{-4}$ were used for all datasets. Our method was evaluated on the validation sets of each dataset, considering both fine-tuning and retraining of the last layer. For all datasets, with the exception of MultiNLI, retraining provided superior validation results. The specifics regarding the number of epochs and the ranges for hyperparameter search (including learning rate, $\ell_1$-regularization coefficient ($\lambda$), and the number of selected samples ($k$)) for each dataset are as follows:

- **Waterbirds**.
    - epochs = 100,
    - lr = $5 \times 10^{-4}$,
    - $\lambda \in \{0, 0.01, 0.02, 0.03, 0.04, 0.05, 0.06, 0.07, 0.08, 0.09, 0.1, 0.2, 0.3, 0.4, 0.5\}$,
    - $k \in \{20, 25, 30, 35, 40, 45, 50, 55, 60\}$.
- **CelebA**
    - epochs = 50,
    - lr = $5 \times 10^{-4}$,

- $\lambda \in \{0, 0.01, 0.02, 0.03, 0.04, 0.05, 0.06, 0.07, 0.08, 0.09, 0.1, 0.2, 0.3, 0.4, 0.5,$
  $0.6, 0.7, 0.8, 0.9, 1, 2\}$,
- $k \in \{50, 100, 150, 200, 250, 300\}$.

- **UrbanCars**
  - epochs = 100,
  - lr $\in \{5 \times 10^{-4}, 10^{-3}\}$,
  - $\lambda \in \{0, 0.01, 0.02, 0.05, 0.1, 1\}$,
  - $k \in \{10, 20, 30, 50, 63\}$.

- **CivilComments**
  - epochs = 50,
  - lr $\in \{10^{-4}, 5 \times 10^{-4}\}$,
  - $\lambda \in \{0, 0.01, 0.02, 0.03, 0.04, 0.05, 0.06, 0.07, 0.08, 0.09, 0.1, 0.2, 0.3, 0.4, 0.5,$
    $0.6, 0.7, 0.8, 0.9, 1, 2\}$,
  - $k \in \{500, 750, 1000, 1250, 1500\}$.

- **MultiNLI**
  - epochs = 200,
  - lr $\in \{10^{-3}, 10^{-2}\}$,
  - $\lambda \in \{0, 0.01, 0.02, 0.03, 0.04, 0.05, 0.06, 0.07, 0.08, 0.09, 0.1, 0.2, 0.3, 0.4, 0.5\}$,
  - $k \in \{20, 30, 40, 50, 60, 75, 100, 125, 150, 200, 250, 300\}$.

- **Dominoes-CMF**
  - `LogisticRegression(penalty="l1", solver="liblinear")`
  - $\lambda \in \{0.001, 0.003, 0.01, 0.02, 0.03, 0.05, 0.07, 0.1, 0.2, 0.3, 0.5, 0.7, 1.0, 3.0\}$,
  - $k \in [10, 80]$.

- **CelebA-SHSG**
  - `LogisticRegression(penalty="l1", solver="liblinear")`
  - $\lambda \in \{0.001, 0.003, 0.01, 0.02, 0.03, 0.05, 0.07, 0.1, 0.2, 0.3, 0.5, 0.7, 1.0, 3.0\}$,
  - $k \in [1, 100]$.

### E.5 SENSITIVITY TO HYPERPARAMETERS

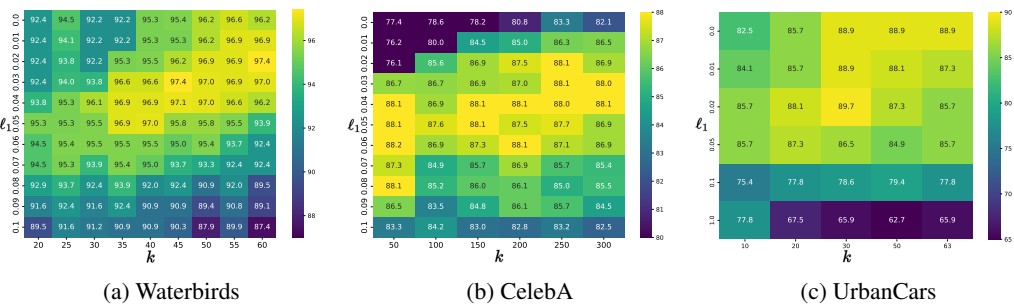

|  (a) Waterbirds  |  (b) CelebA  |  (c) UrbanCars  |

Figure 9: WGA heatmap on $D^{MS}$ for different hyperparameter settings across various datasets.

The parameters $k$ (the number of selected samples from each loss tail) and $\lambda$ (the $\ell_1$ regularization factor) are automatically selected using the environment/group-based validation scheme proposed in our method. Sensitivity heatmaps demonstrate the impact of $k$ and $\lambda$ on the worst-group validation accuracy (WGA) across various datasets. Importantly, our results demonstrate that for most datasets, multiple hyperparameter combinations yield optimal or near-optimal performance, reducing the need for exhaustive searches. This suggests that the hyperparameter tuning process is not prohibitively difficult, and even relatively shallow or targeted hyperparameter searches suffice to identify optimal hyperparameter configurations. The difference in WGA between the best and worst hyperparameter settings for the Waterbirds, CelebA, and UrbanCars datasets is approximately 10%, 16%, and 25%, respectively.

Table 10: Results of DFR and AFR with EIIL-inferred environment for model selection.

| Method | Waterbirds | Celeba |
|---|---|---|
| DFR (with EIIL) | $\mathbf{92.21 \pm 0.02}$ | $\mathbf{85.55 \pm 1.0}$ |
| AFR (with EIIL) | $82.6 \pm 0.04$ | $72.5 \pm 0.01$ |

Table 11: Performance comparison between misclassified sample selection and EVaLS on the Waterbirds, CelebA, and UrbanCars datasets. The mean and standard deviation values are calculated over three runs with different seeds.

| Method | Waterbirds | | CelebA | | UrbanCars | |
|---|---|---|---|---|---|---|
| | Worst | Average | Worst | Average | Worst | Average |
| Misclassified Selection | $77.8_{\pm 5.2}$ | $94.0_{\pm 0.4}$ | $85.9_{\pm 1.0}$ | $89.4_{\pm 0.8}$ | $78.4_{\pm 4.5}$ | $86.9_{\pm 1.4}$ |
| EVaLS | $88.4_{\pm 3.1}$ | $94.1_{\pm 0.1}$ | $85.3_{\pm 0.4}$ | $89.4_{\pm 0.5}$ | $82.1_{\pm 0.9}$ | $88.1_{\pm 0.9}$ |

# F    ABLATION STUDY

## F.1    USE OF EIIL WITH DFR AND AFR

We conducted an ablation study to investigate the impact of using environments inferred from EIIL on model selection. Specifically, we benchmarked the performance of DFR and AFR with EIIL-inferred groups. The results, presented in Table 10, demonstrate the effectiveness of incorporating EIIL-inferred groups in model selection. The results show that while EIIL-inferred groups reduce the performance compared to ground-truth annotations for model selection, they still can be effective for robustness to an extent. Moreover, EVaLS outperforms these two methods when using EIIL inferred environments.

## F.2    COMPARISON OF HIGH-LOSS AND MISCLASSIFIED-SAMPLE SELECTION

Several methods, such as JTT (Liu et al., 2021a), rely on misclassified points to address group imbalances by treating these points as belonging to a minority group. To verify the effectiveness of loss-based sampling in comparison with misclassification-based sample selection, we conducted an experiment by replacing loss-based sampling in in EVaLS with selecting misclassified samples and an equal number of randomly chosen correctly classified samples from each class. This results in degraded performance compared to EVaLS on the Waterbirds and UrbanCars datasets, and only a marginal improvement (with higher variance) on CelebA, as summarized in Table 11.

## F.3    OTHER ENVIRONMENT INFERENCE METHODS

In addition to EIIL, other environment inference methods could be utilized for partitioning the model selection set into environments.

**Error Splitting**    JTT Liu et al. (2021a) partitions data into two correctly classified and misclassified sets based on the predictions of a model trained with ERM. We split each of these two sets based on labels of samples, obtaining $|\mathcal{Y}| \times 2$ environments.

**Random Classifier Splitting**    uses a random classifier to classify features obtained from a model trained with ERM into correctly classified and misclassified sets. Similar to error splitting, we split the sets based on class labels. The difference between error splitting and random classifier splitting is solely in the reinitialization of the classification layer.

The results for EVaLS-ES (EVaLS+Error Sampling) and EVaLS-RC (EVaLS+Random Classifier) are shown in Table 12. One limitation of error splitting is that in datasets with noisy labels or corrupted images, samples that an ERM model misclassifies may not always belong to minority groups. In these situations, choosing models based on their accuracy on corrupted data could lead

Table 12: The performances of three environment inference methods, when combined with loss-based sample selection, are evaluated on spurious correlation benchmarks. The mean and standard deviation values are calculated over three separate runs, each initiated with a different seed.

| Method | Waterbirds | | CelebA | | UrbanCars | |
|---|---|---|---|---|---|---|
| | Worst | Average | Worst | Average | Worst | Average |
| EVaLS-ES | $82.1_{\pm 1.2}$ | $\mathbf{94.3_{\pm 0.04}}$ | $48.4_{\pm 11.6}$ | $69.5_{\pm 6.5}$ | $79.2_{\pm 2.9}$ | $86.1_{\pm 0.9}$ |
| EVaLS-RC | $\mathbf{88.7_{\pm 1.0}}$ | $94.3_{\pm 1.1}$ | $78.1_{\pm 5.1}$ | $\mathbf{93.5_{\pm 0.2}}$ | $\mathbf{82.4_{\pm 3.2}}$ | $\mathbf{88.2_{\pm 0.8}}$ |
| EVaLS | $88.4_{\pm 3.1}$ | $94.1_{\pm 0.1}$ | $\mathbf{85.3_{\pm 0.4}}$ | $89.4_{\pm 0.5}$ | $82.1_{\pm 0.9}$ | $88.1_{\pm 0.9}$ |

to the selection of models that are not robust to spurious correlations. This is demonstrated by the results of EVaLS-ES on the CelebA dataset.

This shortcoming of error splitting can be alleviated by employing a random classifier instead of the ERM-trained one. Due to the feature-level similarity between minority and majority samples in datasets affected by spurious correlation (Sohoni et al., 2020; Kirichenko et al., 2023; Lee et al., 2023), it is expected that the classifier can differentiate between the groups to some extent. As shown in Table 12, surprisingly, EVaLS-RC produces results that are generally comparable to EVaLS. However, the performance of this method may have high variance, depending on the different initializations of the classifier.

## G CELEBA-SHSG DATASET FOR UNKNOWN SPURIOUS CORRELATIONS

To further investigate the performance of EVaLS in scenarios with unknown spurious correlations, we propose the CelebA-SHSG (Straight Hair, Smiling, Gender) dataset. This dataset is a subset of the original CelebA (Liu et al., 2014), where the label *"Straight Hair"* is correlated with the attributes of smiling and being female.

The *"Straight Hair"* attribute is considered as the label, *"Smiling"* as the known spurious attribute, and *gender* as the unknown spurious attribute. Average accuracies and Worst group accuracies (WGA) are reported in Table 13 among 8 groups (all binary combinations of the label and spurious attributes). We set the spurious correlation of the known attribute to $80\%$ and conduct experiments for various levels of unknown spurious correlation (similar to the Dominoes-CMF experiments). Spurious correlations are imposed by subsampling from the original CelebA dataset.

The results demonstrate that methods that do not rely on group annotations for retraining or model selection achieve higher WGA among groups based on both known and unknown attributes. Specifically, EVaLS achieves higher WGA compared to EVaLS-GL, which uses loss-based sampling to create the retraining dataset and relies on group annotations for model selection. Furthermore, EVaLS-GL outperforms DFR, which depends on group annotations for both retraining and model selection. EVaLS improves the WGA of ERM by $22.7\%$ on average. The Oracle model uses group annotations based on both known and unknown spurious attributes during retraining and model selection. Its WGA is $40.8\%$ higher than that of DFR on average, which only uses annotations of the known spurious attribute.

These results further confirm the findings in Figure 4 for Dominoes-CMF.

## H SOCIETAL IMPACTS

Real-world datasets often encapsulate social biases that stem from entrenched stereotypes and historical discrimination, affecting various groups such as genders and races. Machine learning methods, which learn the correlation between patterns in input data and their targets (e.g., labels in a classification task) (Beery et al., 2018), inadvertently absorb this bias. This unintended consequence leads to fairness issues in many applications. While strategies to mitigate such biases have been proposed (as discussed comprehensively in Section A), societal biases are not always known and determined. We believe that our work, as it addresses these unidentified biases, takes a significant step towards making machine learning fairer for our society.

Table 13: A Comparison of ERM, DFR, DFR (Oracle), AFR, AFR+EIIL, EVaLS, and EVaLS-GL on the CelebA-SHSG with different spurious correlations for the unknown feature. Both the worst and average of test group accuracies are presented. The mean and standard deviation are calculated based on runs with three distinct seeds.

| Method | 85% Corr. | | 90% Corr. | | 95% Corr. | |
|---|---|---|---|---|---|---|
| | Worst | Average | Worst | Average | Worst | Average |
| ERM | $28.3_{\pm0.6}$ | $68.2_{\pm0.6}$ | $23.9_{\pm1.5}$ | $67.3_{\pm0.3}$ | $15.6_{\pm2.6}$ | $63.5_{\pm0.8}$ |
| DFR (Oracle) | $63.1_{\pm0.9}$ | $71.7_{\pm1.2}$ | $59.2_{\pm1.9}$ | $70.0_{\pm1.1}$ | $58.4_{\pm5.0}$ | $67.7_{\pm1.5}$ |
| DFR | $27.2_{\pm2.2}$ | $67.7_{\pm0.2}$ | $18.9_{\pm0.7}$ | $64.9_{\pm0.3}$ | $12.3_{\pm1.6}$ | $60.1_{\pm0.3}$ |
| AFR | $28.1_{\pm0.4}$ | $68.0_{\pm0.4}$ | $24.3_{\pm2.1}$ | $65.7_{\pm0.0}$ | $15.7_{\pm2.6}$ | $63.1_{\pm0.0}$ |
| AFR + EIIL | $41.3_{\pm5.7}$ | $63.2_{\pm5.1}$ | $36.3_{\pm4.5}$ | $69.8_{\pm0.0}$ | $45.0_{\pm5.3}$ | $63.2_{\pm0.0}$ |
| EVaLS-GL | $30.5_{\pm5.2}$ | $68.6_{\pm2.3}$ | $26.3_{\pm6.4}$ | $67.4_{\pm1.0}$ | $19.3_{\pm3.2}$ | $61.6_{\pm3.4}$ |
| EVaLS | $\mathbf{45.2_{\pm2.9}}$ | $59.5_{\pm2.7}$ | $\mathbf{44.9_{\pm3.1}}$ | $62.7_{\pm1.8}$ | $\mathbf{45.7_{\pm2.2}}$ | $64.4_{\pm1.8}$ |

# I   COMPUTATIONAL RESOURCES

Each experiment was conducted on one of the following GPUs: NVIDIA H100 with 80G memory, NVIDIA A100 with 80G memory, NVIDIA Titan RTX with 24G memory, Nvidia GeForce RTX 3090 with 24G memory, and NVIDIA GeForce RTX 3080 Ti with 12G memory.

