# OpenReview forum: "Trained Models Tell Us How to Make Them Robust to Spurious Correlation without Group Annotation"
_ICLR.cc/2025/Conference — Submitted to ICLR 2025_

### Official Review · Reviewer_3v9E · 2024-10-31

**Soundness:** 3
**Presentation:** 3
**Contribution:** 3
**Rating:** 8
**Confidence:** 3

**Summary:**

This paper works in the sub-population shift setting, in which the data consists in samples of different groups that shares a property. The goal is to learn a model robust to sub-population shifts, such as class imbalance, attribute imbalance, and spurious correlations. They propose a method (EVaLS) that improves the robustness of models trained using ERM without using any group annotation data. The method is well motivated both empirically and theoretically, and they show that the model has competitive performance with and without using group annotation. Moreover, they show that EVaLS is robust to scenarios where there is an unknown spurious attribute in comparison to the state-of-the-art.

**Strengths:**

- Simple method, that works in multiple scenarios (with and without group annotations).
- The method do not make strong assumptions about the trained model. The only requirement is a model trained by ERM (the training data or any other training information is not necessary), while the method acts in a post-training phase.
- It shows competitive performance in comparison to the literature, while it has a less strict set of requirements in comparison to most of them.
- EVaLS outperforms the DFR (one of the state-of-the-art models) in cases with multiple spurious attributes

**Weaknesses:**

- Make a comparison of EVaLS with more methods (e.g. AFR, since it also doesn't depend on ERM training) in the multiple spurious attributes scenario.
- Have more evidence that EVaLS outperforms the DFR/other methods in cases with multiple spurious attributes (e.g. using more datasets). I believe that this is the strongest part of the results, and it is a clear advantage for EVaLS (besides cases in which group annotation is not available).

**Questions:**

Some questions:
1) Is it feasible to add AFR results to the multiple spurious attributes experiment?
2) Is it feasible to add an extra dataset to the multiple spurious attributes experiment?

Adding these extra results will address the points mentioned in the weakness, showing stronger empirical evidence about EVaLS advantage in cases with multiple spurious attributes.

Minor points:
- There are references to Figure 1 (Line 83) and Figure 3 (Lines 172-177) that come before Figure 2 and were a bit confusing to me while I was reading the paper for the first time. In my opinion, removing these references will improve the readability of the paper.
- Figure 2 instead of figure 2 (Line 205).
- Have the oracle results as a reference in Table 6 to follow Figure 4 (b).
- Figure 4 (b) could include the standard deviation (such as in Table 6).
- Sometimes you use sub-population (Line 131) or subpopulation (Line 139). Keep it consistent across the paper.

---

> ### Author Response · Authors · 2024-11-18
> **Reply to Reviewer 3v9E**
>
> Dear Reviewer 3v9E,
>
> First, we appreciate your acknowledgment of the importance and impact of results that demonstrate robustness in scenarios where some or all spurious correlations are unknown. We believe this is a highly influential step in the field.
>
> ---
>
> ## Questiion 1
>
> WGA of AFR on Dominoes-CMF are as follows:
>
> |Correlation of the unknown spurious feature|AFR|AFR+EIIL|
> |-|-|-|
> |85|$65.7_{\pm0.2}$|$69.1_{\pm0.1}$|
> |90|$54.2_{\pm0.2}$|$61.5_{\pm0.2}$|
> |95|$40.3_{\pm0.5}$|$40.4_{\pm0.1}$|
>
> A similar improvement from EVaLS to EVaLS-GL is observable between AFR adn AFR+EIIL.
>
> ---
>
> ## Question 2
>
> A dataset that has multiple spurious shortcuts, each highly correlated with the target independently, requires a large amount of data to capture both spurious correlations. Constructing such a dataset with proper quality, along with training and evaluating models on it, requires some time to accomplish. We are working on it and will inform you.
>
> ---
>
> ## Minor points
> Thank you for pointing out the slips and other aspects that may cause potential confusion in our paper. We will correct the slips and try to address your suggested improvements in the revision.

---

> ### Comment · Reviewer_3v9E · 2024-11-22
>
> Thank you for the reply. Clearly EVaLS outperforms AFR and AFR+EIIL in the Dominoes-CMF, and it would be a great addition to Figure 4 (b) and Table 6. Is it possible to have them updated in the pdf?
>
> Regarding the dataset, I understand that it is not a trivial task for a rebuttal process.

---

> > ### Author Response · Authors · 2024-11-25
> > **New Dataset**
> >
> > ### Minor Points
> >
> > As you can now see in the revised version, we have added AFR results to Figure 4(b). However, we feel that the figure has become too crowded, reducing its readability. Therefore, we think it might be better to keep the figure as it was before, avoid adding further information and move the additional information to Table 6, which we believe is more readable and preferable. Please let us know if you strongly prefer otherwise.
> >
> > We have corrected the slips in the original submission. Since changes to the body of the paper would invalidate the line references in our responses to other reviewers, we will update Table 6, change the references and/or order of the figures, and add results of the dataset below later in the rebuttal period to avoid confusion for the other reviewers.
> >
> > ---
> >
> > ### New Dataset
> > We have added a new dataset for evaluating the robustness of trained models to unknown shortcuts (Sec. 2.2). We have used a subset of the CelebA dataset. The new dataset consists of real-world images and further confirms our findings on the effectiveness of EVaLS in robustness against known and unknown spurious correlations.
> >
> > The *“Straight Hair”* attribute is considered as the label, *“Smiling”* as the known spurious attribute, and *gender* as the unknown spurious attribute. Worst group accuracies (WGA) are reported in the table below among 8 groups (all binary combinations of the label and spurious attributes). We set the spurious correlation of the known attribute to 80% and conduct experiments for various levels of unknown spurious correlation (similar to the Dominoes-CMF experiments). Spurious correlations are imposed by subsampling from the original CelebA dataset.
> >
> > The results demonstrate that methods that do not rely on group annotations for retraining or model selection achieve higher WGA among groups based on both known and unknown attributes. Specifically, *EVaLS* achieves higher WGA compared to *EVaLS-GL*, which uses loss-based sampling to create the retraining dataset and relies on group annotations for model selection. Furthermore, *EVaLS-GL* outperforms *DFR*, which depends on group annotations for both retraining and model selection. *EVaLS* improves the WGA of ERM by $22.7$% on average. The *Oracle* model uses group annotations based on both known and unknown spurious attributes during retraining and model selection. Its WGA is $40.8$% higher than that of DFR on average, which only uses annotations of the known spurious attribute.
> >
> > These results further confirm the findings in *Figure 6(b)* for Dominoes-CMF.
> >
> > |Method|85% Unknown Spurios Corr.|90% Unknown Spurios Corr.|95% Unknown Spurios Corr.|
> > |-|-|-|-|
> > DFR (Oracle)|$63.1_{\pm 0.9}$|$59.2_{\pm 1.9}$|$58.4_{\pm 5.0}$|
> > DFR|$27.2_{\pm 2.2}$|$18.9_{\pm 0.7}$|$12.3_{\pm 1.6}$|
> > AFR+EIIL|$41.3_{\pm 5.7}$|$36.3_{\pm 4.5}$|$45.0_{\pm 5.3}$|
> > AFR|$28.1_{\pm0.4}$|$24.3_{\pm 2.1}$|$15.7_{\pm2.6}$|
> > EVaLS|$\boldsymbol{45.2}_{\pm 2.9}$|$\boldsymbol{44.9}_{\pm 3.1}$|$\boldsymbol{45.7}_{\pm2.2}$|
> > EVaLS-GL|$30.5_{\pm5.2}$|$26.3_{\pm 6.4}$|$19.3_{\pm 3.2}$|
> > ERM|$28.3_{\pm 0.6}$|$23.9_{\pm 1.5}$|$15.6_{\pm 2.6}$|
> >
> > We have addressed all your questions. If further clarification or information is needed, we are eager to address them promptly.

---

> > > ### Comment · Reviewer_3v9E · 2024-11-25
> > >
> > > Thank you for your answer.
> > >
> > > Yes, Figure 4 (b) has too much information. Comparing the results using Table 6 only seems easier for me.
> > >
> > > Regarding the new dataset, that's indeed a very good result. I am increasing my score accordingly.

---

> > > > ### Author Response · Authors · 2024-11-26
> > > >
> > > > Thank you for your constructive review and consideration of our rebuttal.
> > > >
> > > > Best regards,
> > > >
> > > > Authors of Submission #12227

---

### Official Review · Reviewer_9oxU · 2024-11-01

**Soundness:** 3
**Presentation:** 2
**Contribution:** 2
**Rating:** 5
**Confidence:** 3

**Summary:**

This paper introduces a method called EVaLS, which trains a classifier that is robust to spurious correlations without requiring group labels. The method constructs a balanced dataset based on loss values and utilizes environments for hyperparameter tuning. Experiments on several datasets demonstrate the effectiveness of the proposed method.

**Strengths:**

The method of selecting high-loss and low-loss data proposed in this paper effectively mitigated the problem of group imbalance, with both experiments and theoretical analysis effectively validating this point.

**Weaknesses:**

- The authors may need to provide a more detailed description of the advantages of loss-based sampling over other sampling methods. For instance, it would be beneficial to compare it specifically with methods like SELF, highlighting the unique benefits or efficiencies achieved by the proposed method.

- The authors' claim that their method 'completely eliminates the need for group annotations' as a primary contribution seems somewhat tenuous. The methodology presented in the paper can be categorized into two main components: Environment-based Validation (EV) and Loss-based Sampling (LS). Notably, LS appears to require group labels for hyperparameter tuning, while EV utilizes environment labels generated through Environment Inference for Invariant Learning (EIIL) as a stand-in for group labels. However, EIIL was originally proposed by Creager et al., 2021, and it was inherently designed to address scenarios where group labels are not available. Although Creager et al., 2021 used true group labels for model selection within GroupDRO in their experiments, it appears that environment labels could also be suitably employed for this step.

**Questions:**

- From lines 452-457, the authors state that annotation-free methods can mitigate the impact of both labeled and unlabeled shortcut features more effectively. However, EVaLS-GL, which utilizes group labels, achieved better results than EVaLS. Could the authors provide their insights on this phenomenon?
-
In the experiments, the performance of EVaLS and EVaLS-GL varied, with each having its strengths and weaknesses. Could the authors discuss the advantages and disadvantages of using inferenced environments versus group labels based on these findings?

---

> ### Author Response · Authors · 2024-11-19
> **Reply to Reviewer 9oxU - Part 1/3**
>
> Dear Reviewer 9oxU,
>
> Thank you for your valuable feedback and acknowledgment of the effectiveness of our work. We appreciate your attention to various aspects of the framework. Below, we aim to clarify confusing points and provide a comprehensive discussion and analysis on the questions you have raised.
>
> ---
> ## Weakness 1
> ### Compared to Sampling Method in JTT
> Please refer to Section F.2, *Comparison of High-Loss and Misclassified-Sample Selection*.
>
> Having a hyperparameter ($k$) to control the number of selected samples from high loss samples provides more flexibility to LS compared to JTT. There is a tradeoff between the purity and the number of selected high-loss samples: our observations (Figure 2 (3 in the revision)) demonstrate that minority samples are more commonly found among those with high loss in the ERM model. As the number of selected high-loss samples increases, the proportion of minority samples among them decreases. However, selecting more samples could improve the overall training for those samples. The flexibility to choose among various numbers of selected samples allows EVaLS to find an optimal point in this tradeoff. Sensitivity to the selection of $k$ is shown in Figure 9 of the paper. Choosing only misclassified samples cannot handle this tradeoff effectively, particularly when the number of misclassified samples is either too high or too low. See Table 11 for performance comparison.
>
> ### Compared to SELF
> SELF uses a different criterion for selecting minority samples. It relies on the disagreement between the outputs of a trained model and an early-stopped model, leveraging the observation that models behave differently with respect to spurious correlations during the final and early stages of training. In contrast, we base our approach on the observation that minority samples are more prevalent among high-loss samples than low-loss ones.
> Both approaches yield effective results. However, loss-based sampling (EVaLS-GL) outperforms disagreement-based sample selection by SELF in 4 out of 5 benchmarks, under a similar level of group supervision. Moreover, as stated in L371–372 (390-91 in the revision), SELF requires additional training information (an early-stopped model), whereas loss-based sample selection does not (see methods with $\star$ in Table 1). This makes SELF less feasible in many cases where the model or training data is large, and early-stopped checkpoints are unavailable.

---

> ### Author Response · Authors · 2024-11-19
> **Reply to Reviewer 9oxU - Part 2/3**
>
> Before proceeding, we want to clarify some points that may have been misunderstood:
>
> #### **1. EVaLS-GL, which utilizes group labels, achieved better results than EVaLS**
> This is logical and not a contradiction. For every spurious attribute that its exact group annotations are available, using the annotations to create a balanced (re)training dataset or for model selection is the optimal approach to ensure robustness to correlation shifts in that spurious attribute. This can be observed by comparing DFR and GDRO with other methods in Table 1. In other words, it is not expected that EVaLS, which does not use group-level data, will surpass models that leverage this additional information for robustness to those groups. Nonetheless, as we state in our Abstract (L32–34), the fact that EVaLS achieves **near-optimal** results under these constraints demonstrates its robustness and practical utility, especially since it can be used in scenarios where the aforementioned group annotations are not available.
>
> This property makes EVaLS applicable to scenarios where **the spurious correlations a trained model relies on are unknown** during (re)training. Since no solution before us has solved this problem—prevalent across machine learning models in real-world applications—EVaLS represents a significant advancement in improving model robustness against spurious correlations. **Achieving near-optimal accuracy without group annotations, while also addressing robustness to unknown spurious correlations, makes EVaLS an effective solution for real-world scenarios where some spurious correlations are known and others are not.**
>
> To more effectively illustrate this point, we proposed Dominoes-CMF. In Figure 4(b), you can see that EVaLS achieves higher worst-group accuracy across all 8 groups (as demonstrated in Figure 4(a) and explained in Section 2.2, *Robustness of a Trained Model to Unknown Shortcuts*) compared to EVaLS-GL, which requires group annotations for model selection, and DFR, which uses group annotations for both retraining and model selection.
>
> #### **2. LS appears to require group labels for hyperparameter tuning, while EV utilizes environment labels generated through Environment Inference for Invariant Learning (EIIL) as a stand-in for group labels**
> Environments are not considered equivalent to groups. They are diverse subsets of the data. In our framework, what we require from environments is that they depict group shifts (see Section 3.2, L214–215 (259-260 in the revision) and L244–258 (263-269 and 285-292 in the revision); Abstract, L26–30; Introduction, L73–74 and L77–80). For example, majority samples of a class may be more prevalent in two environments, but this is not problematic as long as there is a notable shift in the percentage of minority samples between them (see Table 2 for group shifts in the obtained environments across datasets).
> Consequently, the performance of EVaLS demonstrates that LS does not necessarily require group labels for hyperparameter tuning. Instead, combining it with worst-environment accuracy, derived from environments with the above properties, can achieve near-optimal accuracy.

---

> ### Author Response · Authors · 2024-11-19
> **Reply to Reviewer 9oxU - Part 3/3**
>
> ## Weakness 2
> First, please refer to point 2 above for clarification. Results for GDRO+EIIL (Creager et al., 2021) are shown in Table 1. It can be seen that it underperforms other methods in the table in most comparisons, indicating that their approach is not as effective as others. Specifically, GDRO+EIIL underperforms GDRO, which utilizes group annotations, as expected. You can find more information about results on spurious correlation datasets below.
>
> ---
>
> ## Question 1
> First, please refer to point 1 above for clarification. When group annotations are available, we expect methods that utilize them to outperform others. So, it is expected that EVaLS-GL outperforms EVaLS when exact group annotations are available.
>
> ---
>
> ## Question 2
> Let’s first review datasets with spurious correlations in Table 1 and 6, and then compare the results of EVaLS and EVaLS-GL.
>
> On *Waterbirds* with available group annotations, EVaLS-GL outperforms EVaLS by $1$%.
>
> Regarding *CelebA*, EVaLS achieves a slightly higher WGA ($0.7$%). It is known (L1449-1453 (1455-1457 and 1470-1471 in the revision)) that annotation noise is present in this dataset, as evidenced also by previous research (e.g., [\*]). Consequently, group annotations in CelebA are not completely accurate. EVaLS bypasses this issue by utilizing inferred environments instead of relying on noisy group annotations.
>
> On datasets with two spurious attributes—one known and one unknown (*UrbanCars* and *Dominoes-CMF*)—utilizing inferred information rather than available annotations provides a solution for fairly achieving robustness to all spurious attributes. Based on the results of both experiments, EVaLS outperforms EVaLS-GL by an average margin of $1.89$%.
>
> ### Conclusion
> If group annotations for a known spurious correlation are available, using them is more effective for robustness to the spurious correlation than utilizing environments. However, environment-based validation can also achieve near-optimal accuracy (L517–518 (520-521 in the revision)). If group annotations are noisy, this may result in lower performance when utilizing them. In such cases, there is a trade-off between using noisy group annotations and inferred environments.
>
> If a spurious correlation that a trained model relies on is unknown during training, using inferred environments is useful for robustness, whereas methods that require explicit annotations are not applicable.
>
> Combining the above points, for model selection:
>
> - **If the goal is to ensure robustness to a known spurious correlation with ground-truth annotations, using group annotations is an effective solution (depending on the quality of the ground-truth annotations).**
> - **If the goal is to generally improve the robustness of a trained model to both known and unknown spurious correlations, using inferred environments is an effective solution.**
>
> [*] Speth et al., *Automated Label Noise Identification for Facial Attribute Recognition*, CVPR, 2019.

---

### Official Review · Reviewer_3i2u · 2024-11-03

**Soundness:** 2
**Presentation:** 3
**Contribution:** 2
**Rating:** 6
**Confidence:** 4

**Summary:**

Many studies that enhance robustness to spurious correlation require group annotations for training. This paper aims to enhance the robustness with minimal group annotation assumptions. Specifically, the losses from an ERM-trained model are used to construct a balanced dataset of high-loss and low-loss samples, mitigating group imbalance in data. Moreover, using environment inference methods to create diverse environments has been shown to potentially eliminate the need for group annotation in model selection. Experiments demonstrate the effectiveness of the proposed method.

**Strengths:**

- The paper is well-written and easy to follow.

- The paper proposes a practical method to mitigate the reliance on spurious correlations without any group annotations.

- A new dataset is constructed that demonstrates the effectiveness of the proposed method in mitigating unknown shortcuts.

**Weaknesses:**

- The proposed method is incremental and has limited technical contributions. Retraining the last-layer using group-balanced validation data to mitigate the reliance on spurious correlations has been used in [1,2]. Inferring environment is a direct follow-up of [3].

- The theoretical analysis does not really explain why loss-based sampling within a class can be used to create a group-balanced dataset. The analysis assumes that the losses on the majority and minority samples follow Gaussian distributions. Under this assumption, it is obvious that the loss-based sampling could create two group-balanced sets of data. However, whether this assumption holds in practice is questionable. Moreover, a previous study [2] has found that that model disagreement may effectively upsample worst-group data, or in other words, may create a more group-balanced dataset. Thus, the loss-based sampling may not be as effective as proved in the paper.

- The comparison in Table 1 isn't fair for some methods. EVaLS uses new data, i.e., a part of validation data, for retraining, while methods including GDRO + EIIL, JTT, and ERM do not have access to the new data. Moreover, the existing work [4] also propose a method that aims to mitigate spurious correlations without group annotations. It would be beneficial to compare with this method under the same setting.

- There are some model selection methods that do not require group annotations, such as minimum class difference [4] and worst-class accuracy [5]. It would be helpful to analyze the effectiveness of the proposed worst environment accuracy in comparison with these techniques.

[1] Kirichenko et al., Last layer re-training is sufficient for robustness to spurious correlations, ICLR 2023.\
[2] LaBonte et al., Towards last-layer retraining for group robustness with fewer annotations, NIPS, 2023.\
[3] Creager et al., Environment inference for invariant learning, ICML, 2021.\
[4] Li et al., Bias Amplification Enhances Minority Group Performance, TMLR, 2024.\
[5] Yang et al., Change is hard: A closer look at subpopulation shift, ICML, 2023.

**Questions:**

- See the weaknesses.
- In Table 1, why the experiments on the CivilComments and MultiNLI datasets are out of the scope of the method?
- In L189, the authors mention that they randomly divide the validation set into $\mathcal{D}^{LL}$ and $\mathcal{D}^{MS}$. What if the random division results in a poor set of $\mathcal{D}^{MS}$ which does not have sufficient samples to represent a minority group of samples?

---

> ### Author Response · Authors · 2024-11-18
> **Reply to Reviewer 3i2u - Part 1/3**
>
> Dear Reviewer 3i2u,
>
> We appreciate your positive feedback on the effectiveness of our scheme and the clarity of our paper.
>
> ---
> ## Weakness 1
> EVaLS is a scheme designed to ensure annotation-free robustness of trained models against spurious correlations (L74–76). In this scheme, modules could be replaced while completely maintaining the overall framework and contribution. As shown in Appendix F, we have conducted comprehensive ablation studies to evaluate these components:
> - Loss-based sampling alternatives: Examined in Sec. F.1 and F.2.
> - Replacing EIIL with other environment inference techniques: Detailed in Sec. F.3.
>
> It is important to clarify that our work has not proposed using or combining previous methods as a novelty. Instead, our contributions lie in demonstrating the following points (L106–120):
> 1. Contrary to prior approaches (Appendix A, L728–750), we show that exact group annotations are not a prerequisite for robustness to spurious correlations, and identifying environments with group shifts (Sec. 3.2) proves effective for model selection.
> 2. When ground-truth annotations are unavailable or spurious correlations are unknonw, leveraging the trained model’s learned representations can effectively enhance robustness to spurious correlations (Figure 4(b) and Table 1—UrbanCars).
> 3. Loss-based sampling (Sec. 3.1) is not only an effective method for ensuring robustness to group shifts (compared to other methods such as those in JTT and SELF; see EVaLS and EVaLS-GL in Table 1), but it is also supported by a theory of data balancing with general assumptions (Sec. 3.3 - Theoretical Analysis).
> ---
> ## Weakness 2
> ### Theoretical Analysis
>
> We **do not** assume that the losses on the majority and minority samples follow Gaussian distributions. As we state in L291-292 (311-312 in the revision), “We assume a general assumption that in feature space (output of $g_\theta$), samples from the minority and majority of a class are derived from Gaussian distributions,” where $g_\theta$ is the feature extractor (L183-184 (208-209 in the revision)). To our knowledge, this is one of the most general assumptions in the theoretical analysis of majority-minority dynamics. It does not even assume different dimensions for core and spurious features (as [6] or [7] do to formulate spurious correlation for other goals and analysis). By this assumption, the Gaussian distribution of our logits (not losses) is derived (Lemma D.1). As mentioned in L294-296 (314-316 in the revision), the order of samples in logit space and loss space is monotonic within each class, and thus the tails are equivalent in these two spaces. Thus, loss-based sampling is backed by our proposed theoretical analysis.
>
> Moreover, please note that, as we state in L285-287 (306-308 in the revision) and as observed in Proposition D.1, it is not obvious and always the case that loss-based sampling could create two group-balanced sets of data. Condition (i) (Eq. (5) in L924) and Condition (ii) (L926-943) are necessary and sufficient conditions for this purpose. Whether these conditions are met depends on the classifier, distance between distributions and their variances in feature space (and consequently in logit space), and the amount of spurious correlation.
>
> Additionally, as stated in L310-311 (330-331 in the revision), we have added a practical justification in Appendix D.2 to show that the conditions of Proposition 3.1 (and more completely in D.1) are met in practice.
>
> ### Comparison to [2]
>
> [2] proposes SELF, which has been compared to our approach in Table 1. It uses a different criterion for sample selection, which, based on results in Table 1, is less effective than our loss-based sampling. More precisely, loss-based sampling (EVaLS-GL) outperforms disagreement-based sample selection by SELF with a similar level of group supervision in 4 of 5 benchmarks. Moreover, As it is stated in L371-372 (390-391 in the revision), SELF requires training information (an early-stopped model), in contrast to our approach and methods like DFR and AFR (indictated by $\star$ in Table 1).
>
> [6] Nagarajan et al., Understanding the failure modes of out-of-distribution, ICLR, 2021.
>
> [7] Sagawa et al., An investigation of why overparameterization exacerbates spurious correlations, PMLR, 2020.

---

> ### Author Response · Authors · 2024-11-18
> **Reply to Reviewer 3i2u - Part 2/3**
>
> ## Weakness 3
>
> Note that for last layer retraining and model selection data, we follow the same scheme as in DFR [1] (Please see their Sec. 6 “Feature Reweighting Improves Robustness,” Hyper-parameters paragraph), which is also used in future works (see Appendix D Training Details in SELF [2]). The important point here is that last layer retraining data is not used for feature learning. It is used for reweighting the features that are learned by the feature extractor $g_\theta$ (L183 (208 in the revision)) during training, which is a relaxed form of feature selection (see Figure 1 in DFR [1]).
>
> Regarding BAM [4], we noticed there are evidences of unreliable results as you can see below.
> Before stating them, note that as we review in Sec. 2. Problem Setting (L139-149), there exist several types of subpopulation shifts with different sources, including class-imbalance (L140-141) and spurious correlation (L142-157). Following Yang et al. (2023b): Given input $x = (x_{c}, x_{s}) \in \mathcal{X}$, which consists of core pattern $x_{c}$ and spurious pattern $x_{s}$, and label $y \in \mathcal{Y}$, we can write the classification model: $$ P(y|x) = \frac{P(x|y)}{P(x)}P(y) = \frac{P(x_{c}, x_{s}|y)}{P(x_{c}, x_{s})}P(y) = \frac{P(x_{c}|y)}{P(x_{c})}\frac{P(x_{s}|x_{c}, y)}{P(x_{s}|x_{c})}P(y) $$ Spurious correlation occurs when $P(x_{s}|x_{c}, y) \gg P(x_{s}|x_{c})$, while class imbalance represents the scenario where $P(y) \gg P(y')$ for $y, y' \in \mathcal{Y}$. Thus, such shifts can occur independently in datasets (see Section 2 Preliminaries L139-149), .
>
> The model selection criterion used by BAM [4], ClassDiff, is a criterion for enhancing robustness to class-imbalance, and it was not reasonable to use it for spurious correlation. To better illustrate why, note that if you have a completely random classifier, w.h.p. it achieves near zero ClassDiff, while its WGA w.h.p. would not be much higher than random. In a more concrete example, setting their auxiliary coefficient ($\lambda$) to 0 and #Epochs in Stage 1 ($T$) to 0, and upweight factor ($\mu$) to 1 in BAM, is equivalent to simply training an ERM model. Using such hyperparameters to train a model on a dataset results in the same WGA as ERM. If the dataset contains a spurious correlation but has no class imbalance, both ClassDiff and WGA will be low.
>
> We reviewed their Appendix B Training details and found that they used a limited range of hyperparameters (see BAM [4] Table 6). By avoiding scenarios that lead to lower (better) ClassDiff but lower (worse) WGA, the results are reported for settings that have a minimum level of bias amplification during initial training and a proper version of the initially trained model.
> Note that the advantage of bias amplification (Nam et al., 2021 in the paper) and using an early-stopped version for identifying spurious correlation (Zhang et al. in the paper, JTT) are known, and it is expected that for a set of hyperparameters the method works. However, the effectiveness of ClassDiff was questionable, and we did not find enough evidence of its effectiveness (because the BAM was tested on a narrow range of hyperparameters). To further justify our thoughts, for UrbanCars, we tested BAM [4] on a set of hyperparameters outside the reported range, and in comparison to in-range hyperparameters, we observed that while the ClassDiff became lower, WGA did not show improvement compared to that of ERM.
>
> Thus, we concluded that there is not enough evidence of effectiveness to benchmark BAM [4] and decided to exclude it from the comparison.
>
> ---
>
> ##  Weakness 4
> The worst group accuracy and average accuracy (in parentheses) results you have requested are as follows:
>
> |model selection criteria|Waterbirds|CelebA|UrbanCars|
> |-|-|-|-|
> |minimum class difference [4]|$80.7_{\pm4.1}$($90.1_{\pm0.2}$)|$75.0_{\pm2.8}$($92.9_{\pm0.4}$)|$82.1_{\pm0.5}$($88.0_{\pm0.6}$)|
> |worst-class accuracy [5]|$89.1_{\pm1.0}$($95.3_{\pm0.2}$)|$71.3_{\pm5.5}$($93.6_{\pm0.4}$)|$81.6_{\pm0.8}$($88.2_{\pm0.7}$)|
>
> By comparing EVaLS results in Table 1 and Table 4, we can see that in most setups, both criteria underperform compared to EV. *Worst-class accuracy* achieves a $0.7$% higher WGA on Waterbirds and *minimum class difference* shows similar WGA but lower average accuracy on UrbanCars compared to EV. On CelebA, both criteria underperform EV by a margin of more than $10$%. *Minimum class difference* results in a $7.7$% lower WGA on Waterbirds, and *worst-class accuracy* underperforms EV by $0.5$% on UrbanCars.

---

> ### Author Response · Authors · 2024-11-19
> **Reply to Reviewer 3i2u - Part 3/3**
>
> ## Question 2
>
> As stated in Sec. 2.1 (L139-L149), there exist multiple types of subpopulation shifts (see response to weakness 4). Refer to Yang et al. (2023b) in the paper for a categorization of subpopulation shifts and their formulation (Table 1 in their paper). EVaLS is designed for robustness to *spurious correlation*, but CivilComments and MultiNLI are examples of other forms of subpopulation shifts (class imbalance and attribute imbalance). It is stated in Abstract L100-101, Experiments L348-350 (368-370 in the revision), and Appendix E.3 L1274-1276 (1277-1279 in the revision), and reported also by Yang et al. (2023b) (see Table 2 in their paper).
>
> As explained in L458-466, patterns distinguishing groups in CivilComments and MultiNLI are **not predictive for the target** class (see tables below). This reduces their visibility in the model’s final layers (see Lee et al. (2023) for the role of various layers of neural networks in different types of shifts). Since environment inference algorithms like EIIL (see Appendix F.3 for further investigation) depend on the last layers of a trained model, they cannot infer environments with notable group shifts (defined in L253-254 (257-259 in the revision)) in CivilComments and MultiNLI. The group shifts in CivilComments and MultiNLI (L464-466 (467-469 in the revision)) are significantly lower than those of the datasets reported in Table 2. Thus, the focus of environment-based validation is on datasets with spurious correlations (see limitations (L528-529 (530-531 in the revision)) and future works (L534-535 (537-539 in the revision)) in the Discussion section).
>
> Nevertheless, loss-based sampling (LS) is effective for CivilComments and MultiNLI. Our EVaLS-GL, using ground-truth group labels for model selection and loss-based sampling for retraining, outperforms other methods with a similar level of group supervision on MultiNLI. Also, its WGA on CivilComments is also only $2.1$% lower than the state-of-the-art method (by DFR [1]).
>
> **Proportion of attributes in each class for CivilComments dataset (Table 9 in the paper)**
> |Toxicity (Class)|Male|Female|LGBTQ|Christian|Muslim|Other Religions|Black|White|
> |-|-|-|-|-|-|-|-|-|
> |0|0.11|0.12|0.03|0.10|0.05|0.02|0.03|0.05|
> |1|0.14|0.15|0.08|0.08|0.10|0.03|0.10|0.14|
>
> **MultiNLI training sets statistics**
> |Group|Class (entailment)|Attribute|# Train Data|
> |-|:-:|:-:|:-:|
> |$G_1$|0|No Negations|57498 (28%)|
> |$G_2$|0|Has Negations|11158 (5%)|
> |$G_3$|1|No Negations|67376 (32%)|
> |$G_4$|1|Has Negations|1521 (1%)|
> |$G_5$|2|No Negations|66630 (32%)|
> |$G_6$|2|Has Negations|1992 (1%)|
>
> ---
>
> ## Question 3
> Note that for last-layer retraining and model selection data, we follow the same scheme as in DFR [1] (please see their Sec. 6, “Feature Reweighting Improves Robustness,” Hyper-parameters paragraph), which is also used in other works (see Appendix D: Training Details in SELF [2]). If the number of minority samples $n$ is larger than ~20 in the validation set, which is the case in all the datasets we have, the number of samples in $D_{MS}$ can be approximated by a normal distribution $\mathcal{N}(\mu = \frac{n}{2}, \sigma^2 = \frac{n}{4})$. By calculating a confidence interval, the probability that fewer than $\frac{n}{4}$ of the samples are in $D_{MS}$ or $D_{LL}$ is approximately $2\Phi(-\frac{\sqrt{n}}{2})$, which would be very low. Thus, like every other case in machine learning, and specifically in our datasets and settings similar to DFR [1], random splits work properly and place an acceptable percentage of samples in each of $D_{MS}$ and $D_{LL}$ with a probability of almost 1. However, in the rare cases you described or other cases with a limited number of data points (as we state in Section 5: Discussion, L526-528 (529-530 in the revision), as a limitation), EVaLS will struggle if not enough minority samples exist in their retraining and model selection dataset.

---

### Official Review · Reviewer_jr76 · 2024-11-06

**Soundness:** 2
**Presentation:** 2
**Contribution:** 3
**Rating:** 5
**Confidence:** 3

**Summary:**

The paper EVaLS, a method to improve model robustness against spurious correlations without requiring group annotations. EVaLS balances high- and low-loss samples from an ERM-trained model and applies a simple last-layer retraining (on a loss-based sampled dataset), thus enhancing group robustness. The approach also uses worst environment accuracy to for model selection. Experimental results on diverse dataset shows competitive performance to baseline methods.

**Strengths:**

1. The proposed approach EValS, using environment inference and Loss-based sampling, is novel and interesting.
2. EValS has competitive performance with other methods across diverse datasets.

**Weaknesses:**

1. The assumption that "minority samples are more prevalent among high-loss samples, while majority samples dominate the low-loss category" is questionable. It is easy to construct distributions that does not satisfy this assumption.
2. The performance of EValS seems to rely on the EIIL to find the correct environment. However, how to find the environments may be a challenging problem itself.
3. The peformance of EValS does not seem consistently better than other baseline methods across all datasets. It is not clear whether the better performance in specific dataset is by chance.

**Questions:**

See weakness above.

---

> ### Author Response · Authors · 2024-11-18
> **Reply to Reviewer jr76 - Part 1/2**
>
> Dear Reviewer jr76,
>
> We appreciate the feedback and the opportunity to clarify our work's contributions. We are delighted that you find our scheme novel and interesting.
>
> ---
>
> ## Weakness 1: Issue with High-Loss Minority Samples
>
> Note that EVaLs aims to enhance a trained model's robustness against **spurious correlations it relies on** (L523-524 (525-526 in the revision)). For a grouping based on an attribute \(a\), if the loss distribution for groups within a class—distinguished by the presence or absence of attribute \(a\)—does not reflect distinct populations, it indicates that the model does not rely on attribute \(a\) for its predictions.
>
> The loss distribution of a trained model across different groups has **certain properties**. We would like to provide additional clarification and evidence on this matter:
> 1. **Our Empirical Observations**:
> As stated in Section 3.1 (L205-208 (249-252 in the revision)), our observations (Figure 2 (3 in the revision)) strongly support the assumption that minority samples are overrepresented among high-loss examples.
> 2. **Loss Dynamics Across Minority and Majority Groups**:
> In the majority groups, both the core and spurious patterns are aligned and predictive of the target. In these scenarios, both patterns lead to lower loss. Conversely, in minority groups, core and spurious patterns exhibit contradictory signs for predicting labels. As a result, if the model relies on spurious correlations, the loss for minority groups will be higher.
> 3. **Empirical Risk Minimization (ERM) Principles**:
> In an ERM framework, the learning process prioritizes minimizing the *overall loss* across the dataset. In the context of spurious correlation (Sec 2.1, L143-L156), *majority* groups dominate the data and contribute more the overall loss. Thus, a model which is trained on the data with ERM shoud exhibit lower loss on these groups. Conversely, *minority* groups are *underrepresented*, may incur higher losses. If a majority group consistently exhibits high loss, it suggests a failure in the learning process to converge, contradicting ERM principles.
> 4. **Evidence in Literature**:
> As stated in L61-63, “the loss value of the model, or its alternatives, are popular signals for recognizing minority groups,” which has been utilized in previous works (Liu et al., 2021a; Qiu et al., 2023; Nam et al., 2020; Noohdani et al., 2024).
> ---
>
> ## Weakness 2: Environment Inference:
>
>
> As detailed in Appendix F.3, EVaLS could be used with other environment inference methods. As we stated in Section 1 (Introduction, L81-83) and demonstrated in Table 12, EVaLS-RC, which employs a random linear classifier for environment creation, outperforms EIIL on some datasets. These results further highlight that the effectiveness of EVaLS does not depend exclusively on EIIL.
>
> What EVaLS requires for model selection is a set of diverse environments that exhibit correlation shift (Abstract, L26-30; Introduction, L73-74 and L77-80; Sec 3.2, L214-215 (259-260 in the revision) and L244-258 (263-269 and 285-292 in the revision)). As discussed in L244-258 (263-269 and 285-292 in the revision) and analyzed in Table 2 of the main paper, modest group shifts in the environments are sufficient for EVaLS to perform well. It does not rely on finding "perfect" environments.
>
> Also, as discussed in Appendix A (Related Work, L722-727), environment inference is a different area of research and a broader problem addressed in invariant learning (L713-727).
>
> We should emphasize that EVaLS is a group annotation-free framework that its performance across various inference methods underscores its flexibility.
>
> ---

---

> ### Author Response · Authors · 2024-11-18
> **Reply to Reviewer jr76 - Part 2/2**
>
> ## Weakness 3: EVaLS Consistency:
>
> Note that EVaLS does not utilize group annotations information regarding spurious correlation for model selection or (re)training, and thus, as we clarify in our Abstract L32-34, its **near-optimal** accuracy showcases its effectiveness. This is a significant advancement, particularly in scenarios where group labels are unavailable or unreliable (a common and challenging real-world scenario).
>
> In other words, it is not expected that EVaLS, which does not utilize group-level data, will surpass all models that use this additional information. Nonetheless, the fact that EVaLS achieves comparable results under these constraints demonstrates its robustness and practical utility.
>
> As you can see in Table 1, when group annotations are available for model selection, EVaLS-GL outperforms other methods with a similar level of group supervision on most datasets. However, it underperforms but achieves near-optimal worst group accuracy compared to DFR and GDRO with higher levels of group supervision in most datasets, as expected. It also outperforms DFR and GDRO in the case of the UrbanCars dataset, where annotations for one of its spurious attributes are not available during training (see L454-457 (457-460 in the revision)).
>
> Finally, the claim that EVaLS's better performance on certain datasets is due to chance contradicts our findings. Across diverse datasets with spurious correlations, EVaLS consistently performs well, supported by statistical measures to minimize random variance. Its comparable performance to methods utilizing group information underscores its strength.

---

### Author Response · Authors · 2024-11-30
**General Response and Final Reminder**

Dear Reviewers,


Thank you for your time reviewing our work. **Only 3 days remain** until the end of the discussion period. We have addressed all your concerns and questions. Please read them and let us know if you feel further clarification, justification, evidence, or experiments are needed.

We uploaded a new version of the paper by the ICLR deadline to correct minor errors and include new experiments reported and requested by Reviewer 3v9E. Please review our response to Reviewer 3v9E and Appendix G in the revised paper for the results of the new dataset added to the paper. The line references in our responses have been updated to reflect the revision.

Also, please read the following for a review and clarification of our contribution:

In continuation of previous efforts in the field for robustness to spurious correlations without group annotations (L57-69, L738-751), requiring group annotations for model selection remains a limitation (L67-69, L749-751).

We have found that the availability of a set of disjoint samples (i.e., environments) of data depicting group shifts can effectively serve this purpose, using worst environment accuracy as a reliable surrogate for model selection for robustness to spurious correlations. This is particularly important as there are known (e.g., using EIIL) or very simple (e.g., applying a random linear classifier on top of the feature space of the model) ways to obtain such sets for datasets with spurious correlations. However, please note that inferring environments is a different field of research (L722-727). In other words, what we have shown is that the availability of group annotations can be effectively relaxed to a condition that is possible to satisfy using current knowledge (through environment inference methods, simply applying a random linear classifier on the feature space, or other methods — see Appendix F.3).

We have also proposed an enhanced sampling technique, *loss-based sampling*, which is shown to be more effective than other group balancing schemes for robustness to subpopulation shift (see Table 1 and Table 11 for comparisons of EVaLS-GL with methods at a similar level of group supervision, such as AFR and SELF, and Appendix F.2 for another comparison with the sampling method in JTT). We have backed our balancing scheme with practical observations and theoretical analysis to provide better insights into why it could be effective.

It is expected that approaches that do not use group annotations will underperform compared to those that use group-level information (compare results of double-ticked methods with others in Table 1). In other words, when group annotations based on a spurious attribute are available, using them is logically more effective for achieving robustness to that attribute than not using them. However, *“in many real-world applications, the process of labeling samples according to their respective groups can be prohibitively expensive and sometimes impractical”* (L67-69). EVaLS (Environment-based Validation and Loss-based Sampling), without using group annotations, demonstrates **near-optimal** performance on spurious correlation datasets (L32-33, results in Table 1). While this represents important progress in the field, another property arises in EVaLS that is even more significant.

The ability to achieve robustness to spurious correlations without group annotations enables EVaLS to improve robust to ***unknown spurious correlations***. This means EVaLS can effectively make trained models robust to spurious correlations they rely on, whether identified or unidentified. Without this capability, even if a model becomes robust to a known spurious correlation using current approaches, a **persistent concern** remains about the presence of unknown spurious correlations. Such correlations may affect the model’s predictions and remain undetected, posing significant performance and safety risks. As discussed in Figure 4(b) and its corresponding explanations, as well as in our discussion with Reviewer 3v9E (Appendix G in the revision, https://openreview.net/forum?id=8DuJ5FK2fa&noteId=Eh8QvMMS7y), previous methods could not be responsible for achieving robustness to *unknown* spurious correlations. In contrast, EVaLS improves robustness to both known and unknown attributes.

In other words, when a spurious correlation is unknown, methods requiring group annotations for robustness are not applicable. However, EVaLS, which demonstrates effectiveness in enhancing robustness to spurious correlations without group annotations, improves robustness to all known and unknown spurious attributes and achieves higher worst-group accuracies among all groups (full combinations of core and spurious attributes).

---

Please let us know if you have any other questions. We try to answer them in the remaining time promptly.

---

### Meta-Review · Area_Chair_yFVJ · 2024-12-16

**Metareview:**

The paper having been borderline during the review process, I have read it and made my own idea of its content.

First, the paper makes the repeated claim, also repeated here in rebuttals, that its approach is near-optimal. But there is absolutely *no* formal result in the paper that allow to state such near optimality (certainly not Proposition 3.1, see below). The paper and rebuttal clearly attach to this claim of near-optimality the results of one *experimental* table (1). Such a claim could eventually be supported *iff* the optimal mark was known for the domains (e.g. if the domains had been simulated, with clear evidence or a proof of the best result). This is clearly not the case here and the authors insistence on this so-called near-optimality give the opposite results of the one intended: it ends up overselling results otherwise fine.

It is important to insist at this point that absolutely *nothing* in the paper supports *any* claim of near-optimality of the method proposed.

Then, regarding the theory, there is little in the paper that serves the authors claim of generality. Proposition 3.1 is unfortunately poorly formulated and it is hard to understand the meaning of (1), which does not appear in a sentence (reading the appendix, I suspect it is the result of an unfortunate cut in the paper but any fix I can imagine certainly does not bring any substantial part of the generality claimed). The background of Proposition 3.1 is however *extremely* specific and shallow and *certainly* cannot serve as an illustration of the paper's claims about being of broad appeal (weakness #2 of reviewer 3i2u).

I can only stress the importance of *either* a strong formal analysis of the problem, *or* a more thorough experimental section -- indeed, the range of the key values of the approach are so different between domains (Section E.4) that it seems extremely complicated to figure out a simple rule of thumb to give a new user as a starting point to use the technique. In this context, writing "our results demonstrate that for most datasets, multiple hyperparameter combinations yield optimal or near-optimal performance, reducing the need for exhaustive searches" (L1399) makes little sense for *two reasons*, not just for the near-optimality claim (see above), but also because it is clear that the authors have spent a lot of time doing **domain dependent choices** for the parameters used without much details on why / how choices were made.

In fine, I believe the paper starts with an interesting idea -- indeed, a loss can often be viewed as a negative log-likelihood and so the idea of loss sampling can make sense --, but fails to give the proofs that the idea developed justifies the extremely strong claims made here and there in the paper. I can only recommend that the authors redraft the paper, dig in the theory and orient new experiments to show the approach is simple to use and its (hyper)parameters are at least reasonably easy to figure out even for just a reasonable range.

**Additional Comments On Reviewer Discussion:**

The reviewers gave the authors the opportunity to justify some bold claims made in the paper, but the rebuttal(s) essentially failed to provide substance (apart from the authors insisting further on (near)optimality claims yet without any additional part), see in particular 3i2u and 9oxU.

---

### Decision · Program_Chairs · 2025-01-22

Reject